# An adaptive method for speeding up the numerical integration of chemical mechanisms in atmospheric chemistry models: application to GEOS-Chem version 12.0.0

Lu Shen[1], Daniel J. Jacob[1], Mauricio Santillana[2, 3], Xuan Wang[1], Wei Chen[4]

[1]John A. Paulson School of Engineering and Applied Sciences, Harvard University, Cambridge, MA, USA
[2]Computational Health Informatics Program, Boston Children's Hospital, Boston, MA, USA
[3]Department of Pediatrics, Harvard Medical School, Boston, MA, USA
[4]Department of Physics, Harvard University, Cambridge, MA, USA

*Correspondence to*: Lu Shen (lshen@fas.harvard.edu)

**Abstract.** The major computational bottleneck in atmospheric chemistry models is the numerical integration of the stiff coupled system of kinetic equations describing the chemical evolution of the system as defined by the model chemical mechanism (typically over 100 coupled species). We present an adaptive method to greatly reduce the computational cost of that numerical integration in global 3-D models while maintaining high accuracy. Most of the atmosphere does not in fact require solving for the full chemical complexity of the mechanism, so considerable simplification is possible if one can recognize the dynamic continuum of chemical complexity required across the atmospheric domain. We do this by constructing a limited set of reduced chemical mechanisms (chemical regimes) to cover the range of atmospheric conditions, and then pick locally and on the fly which mechanism to use for a given gridbox and time step on the basis of computed production and loss rates for individual species. Application to the GEOS-Chem global 3-D model for oxidant-aerosol chemistry in the troposphere and stratosphere (full mechanism of 228 species) is presented. We show that 20 chemical regimes can largely encompass the range of conditions encountered in the model. Results from a 2-year GEOS-Chem simulation shows that our method can reduce the computational cost of chemical integration by 30-40% while maintaining accuracy better than 1% and with no error growth. Our method retains the full complexity of the original chemical mechanism where it is needed, provides the same model output diagnostics (species production and loss rates, reaction rates) as the full mechanism, and can accommodate changes in the chemical mechanism or in model resolution without having to reconstruct the chemical regimes.

## 1 Introduction

Accurate representation of atmospheric chemistry is of central importance for air quality and Earth system models (National Research Council, 2016). This is a computationally difficult problem since chemical mechanisms typically include hundreds of species coupled through production and loss pathways, and with lifetimes ranging from less than a second to many years.

Computing the kinetic temporal evolution of such systems involves solving a stiff system of $N$ coupled non-linear ordinary differential equations (ODEs) of the form

$$\frac{d\boldsymbol{n}_i}{dt} = P_i(\boldsymbol{n}) - L_i(\boldsymbol{n}) \qquad (1)$$

where $\boldsymbol{n} = (n_1, \ldots n_K)^T$ is the vector of species concentrations, expressed typically as number densities (e.g., molecules cm$^{-3}$), and $K$ is the number of species in the mechanism. $P_i(\boldsymbol{n})$ and $L_i(\boldsymbol{n})$ are the production and loss rates of species $i$ that depend on the concentrations of other species in the mechanism. Finite-difference solution of the coupled system of ODEs requires an implicit scheme to avoid limitation of the time step by the shortest lifetime in the system (Brasseur and Jacob, 2017). Implicit schemes involve repeated construction and inversion of the Jacobian matrix ($K \times K$) for the system, and this is computationally expensive for large $K$. But the full coupled chemical mechanism may not be needed everywhere in the model domain. For example, highly reactive volatile organic compounds (VOCs) have little influence far away from their source regions. Here we show that we can obtain a substantial reduction of computational cost in a global 3-D model by adaptively adjusting the ensemble of species that actually need to be solved as a coupled system in a given model gridbox. We do so with a general algorithm that is readily applicable to any chemical mechanism or numerical solver.

As the simplest example of an implicit scheme, consider the first-order method which approximates Eq. (1) as

$$f_i(\mathbf{n}(t+\Delta t)) = n_i(t+\Delta t) - n_i(t) - s_i(\mathbf{n}(t+\Delta t))\Delta t = 0 \qquad (2)$$

where $\Delta t$ is the time step and $s_i(\boldsymbol{n}(t + \Delta t)) = P_i(\boldsymbol{n}(t + \Delta t))$ **-** $L_i(\boldsymbol{n}\ (t + \Delta t))$ is the net source evaluated at the end of the time step. This defines a vector function $\boldsymbol{f} = (f_1, \ldots f_K)^T$ and an algebraic system $\boldsymbol{f}(\boldsymbol{n}(t + \Delta t)) = \mathbf{0}$ that is solved iteratively by the Newton-Raphson method. The procedure involves iterative calculation and inversion of the $K \times K$ Jacobian matrix $\boldsymbol{J} = \partial\boldsymbol{s}/\partial\boldsymbol{n}$. Most models use higher-order implicit algorithms designed for accuracy and speed, such as the Gear (Gear 1971; Hindmarsh, 1983) and Rosenbrock (Sandu et al., 1997; Hairer and Wanner, 1991) solvers, but all require iteratively calculating the Jacobian matrix and solving the linear system using a matrix factorization. As a result, the chemical operator that solves for the chemical evolution of species concentrations from Eq. (1) is the most expensive component of atmospheric chemistry models (Eastham et al., 2018), and this computational cost has been a barrier for inclusion of atmospheric chemistry in Earth system models (National Research Council, 2012).

There are various ways to speed up the chemical operator, all involving some loss of accuracy or generality (Brasseur and Jacob, 2017). A general approach is to reduce the dimension of the coupled system of ODEs that needs to be solved implicitly. This can be done by simplifying the chemical mechanism to decrease the number of species (Brown-Steiner et al., 2018; Sportisse and Djouad, 2000), or by isolating long-lived species for which a fast explicit solution scheme is acceptable (Young and Boris, 1977). Jacobson (1995) used different subsets of their full mechanism to simulate the urban atmosphere,

the troposphere, and the stratosphere.

Santillana et al. (2010) combined these ideas in an adaptive algorithm for 3-D models that determines locally at each time step ("on the fly") which species in the chemical mechanism need to be solved in the coupled implicit system. This was done by computing the local production ($P_i$) and loss rates ($L_i$) for all species at the beginning of the time step. Species with either $P_i$ or $L_i$ above a given threshold were labeled "fast" and solved with an implicit scheme, while the others were labeled "slow" and solved with an explicit scheme. The complexity of the chemical system to be solved was thus adapted to the local environment. Here 'fast' and 'slow' refer to the rates in the chemical system, not the species lifetime. For example, short-lived VOCs may be considered slow outside of their source regions because they have negligible influence on other species. The extent of this influence depends on the changing local conditions, hence the need for an adaptive algorithm, The adaptive approach does not pre-judge the local environment, unlike in Jacobson (1995), and instead resolves the dynamic continuum of complexity that may be encountered in the atmosphere. Santillana et al. (2010) applied their algorithm to the GEOS-Chem global 3-D Eulerian chemical transport model (Bey et al., 2001). While the computational savings were promising for the chemical integration within each gridbox, the need to construct a different system in every single grid box and at every time step cancelled out some of the gains and led to only small time-savings when compared to the performance of the standard full-chemistry model.

Here we draw from the approach introduced by Santillana et al. (2010) but use a set of pre-defined chemical regimes to take full advantage of the time-savings of adaptive reduction mechanism algorithm. We start with the objective identification of a limited number of chemical regimes that encompass the range of atmospheric conditions encountered in the model. These regimes are defined by the subset of fast species from the full mechanism that need to be considered in the coupled system, and we pre-code the Jacobian matrix and its inverse for each. Then, the model adaptively picks the appropriate chemical regime to be solved locally and on the fly. We show that this approach can achieve large computational savings without significantly compromising accuracy when implemented in GEOS-Chem. Our method can be adapted to any mechanism and model, retains the complexity of the full mechanism where it is needed, and preserves full diagnostic information on chemical evolution (such as reaction rates, production and loss of individual species, etc.).

## 2 The chemical operator in a 3-D atmospheric model

The evolution of a chemical system within a 3-D Eulerian atmospheric model is obtained by solving the system of $K$ coupled continuity equations (Brasseur and Jacob, 2017)

$$\frac{\partial n_i}{\partial t} = -\boldsymbol{U} \cdot \nabla n_i + P_i - L_i \quad i = [1, K] \qquad (3)$$

where $\boldsymbol{U}$ is the wind vector. The first term on the right-hand side describes the transport of the species, and the other terms describe chemical production and loss as given by Eq. (1). Emission and deposition may be treated as boundary conditions or added to the chemical production and loss terms. Turbulence not resolved by the wind vector can be represented with additional parameterized transport terms.

The system of Eq. (3) must be solved by operator splitting, in which transport and chemistry are solved separately and successively over discrete time steps (Santillana et al., 2016). Operator splitting reduces the dimensionality of the problem because the transport operators do not involve coupling between chemical species, while the chemical operator does not involve spatial coupling. Thus the chemical operator solves the system of ODEs described by Eq. (1) over the operator splitting time step $\Delta t$, passes the updated chemical concentrations to the transport operators, which in turn update the concentrations. After the transport operators have been applied, the results are returned to the chemical operator as initial conditions for the next time step.

The chemical operator is the most computationally costly component of an "off-line" 3-D atmospheric chemistry model where winds and turbulence parameters are taken as input. It remains a major computational burden even in "on-line" models that solve for atmospheric dynamics as well as chemistry. This holds in massively parallel computing environments despite the near-perfect scalability of the chemical operator. For example, Eastham et al. (2018) found in the off-line GEOS-Chem High Performance model at global cubed-sphere c180 ($\approx 50\times 50$ $km^2$) resolution that the chemical operator was responsible for 50% of the overall wall-time using 90 cores and 37% using 360 cores.

Here we will use the off-line GEOS-Chem 12.0.0 global 3-D model for tropospheric and stratospheric chemistry (https://doi.org/10.5281/zenodo.1343547) as demonstration for our algorithm. The model is applied with a horizontal resolution of 4°×5° and 72 pressure levels extending from surface to 0.01 hPa. It is driven by MERRA2 assimilated meteorology from the NASA Global Modeling and Assimilation System (GMAO). The full chemistry mechanism in the model has 228 species and 724 reactions. Among these species, 143 are volatile organic compounds (VOCs), including directly emitted VOCs and their oxidation products, 37 are inorganic reactive halogen species, 24 are organic halogen species, and 24 are other inorganic and aerosol species. The model includes coupled gas-phase and aerosol chemistry in the troposphere and stratosphere as described by Sherwen et al. (2016) and Travis et al. (2016) for the troposphere and Eastham et al. (2014) for the stratosphere. The chemical reactions are integrated using the Rosenbrock solver (Sandu et al., 1997; Hairer and Wanner, 1991) generated from the Kinetic PreProcessor 2.2.4 (KPP) (Damian et al., 2002) software. The model uses operator splitting between chemistry and transport with a chemistry timestep of 20 minutes (Philip et al 2016). We use 12 cores in the simulation.

The key processes in the KPP chemical operator are as follows. The operator first updates the reaction rate coefficients on the basis of temperature, actinic flux, etc. It then passes these reaction rate coefficients together with initial species

concentrations to the Rosenbrock solver, which solves for the temporal evolution of concentrations over the external timestep $\Delta t$. In the process, the Rosenbrock solver approximates the solution at multiple internal timesteps, so it needs to repeatedly recompute the species production and loss rates, construct the corresponding Jacobian matrix, and solve the linear system numerically using a matrix factorization. The bulk of the cost in the overall chemical operator is in the repeated computation of production/loss rates and solving the linear system using a matrix factorization. Reducing the number of species in the system to be solved can significantly reduce the computational cost.

## 3 Adaptive algorithm for the chemical operator

Our adaptive algorithm determines locally what degree of complexity is needed in the chemical mechanism by diagnosing all species in the full chemical mechanism as either "fast" or "slow", and choosing among pre-constructed chemical mechanism subsets ("chemical regimes") which is most appropriate for the local conditions. Here we present (1) the definition of fast and slow species and the different treatments for each, and (2) the approach used to pre-construct the chemical regimes.

### 3.1 Definition of fast and slow species

Following Santillana et al. (2010), we separate atmospheric species as fast or slow based on their production and loss rates in Eq. (1) relative to a threshold $\delta$: fast if either $P_i(\boldsymbol{n}) \geq \delta$ or $L_i(\boldsymbol{n}) \geq \delta$, slow if $P_i(\boldsymbol{n}) < \delta$ and $L_i(\boldsymbol{n}) < \delta$. Concentrations of the fast species are integrated as a coupled system with the KPP Rosenbrock solver. Concentrations of slow species are integrated by explicit analytical solution of Eq. (1) assuming first-order loss with effective rate coefficient $k_i = L_i/n_i$:.

$$\frac{dn_i}{dt} = P_i - k_i n_i \quad (4)$$

$$n_i(t+\Delta t) = \frac{P_i(t)}{k_i(t)} + \left(n_i(t) - \frac{P_i(t)}{k_i(t)}\right)e^{-k_i(t)\Delta t} \quad (5)$$

Solving for $n_i(t + \Delta t)$ by Eq. (5) incurs negligible computational cost, therefore there is considerable advantage in classifying species as slow if this can be done without significant loss in accuracy. We select the threshold $\delta$ for species to be classified as fast or slow by numerical testing, as described in Section 4, but some basic chemical reasoning is useful. Consider the OH radical, which is a central species in atmospheric chemistry mechanisms. OH has a daytime concentration of the order of $10^6$ molecules cm$^{-3}$ and a lifetime of the order of a second, implying production and loss rates of the order of $10^6$ molecules cm$^{-3}$ s$^{-1}$. Species with production and loss rates that are orders of magnitude lower than $10^6$ molecules cm$^{-3}$ s$^{-1}$ are therefore unlikely to influence OH or other species in the coupled mechanism, as these are all to some extent related to OH at least in

the daytime. So we may expect an appropriate threshold $\delta$ to be in the range $10^2$-$10^3$ molecules cm$^{-3}$ s$^{-1}$. Santillana et al. (2010) recommended $\delta$ = 100 molecules cm$^{-3}$ s$^{-1}$ in their algorithm.

One issue with the solution for the slow species by Eq. (5) is that it does not strictly conserve mass, because the loss rate for a given species over the time step does not necessarily match the production rate of the product species. This is usually inconsequential, but we found in early testing that it resulted in the total mass of reactive halogen species growing slowly over time in the stratosphere. To avoid this effect, we treat all 37 reactive inorganic halogen species as fast above 10 km altitude.

### 3.2 Pre-selecting the chemical regimes

Instead of building a local chemical mechanism subset at every time step as in Santillana et al. (210), we greatly improve the computational efficiency by pre-selecting a limited number ($M$) of chemical mechanism subsets (chemical regimes) for which we pre-define the Jacobian matrix in KPP. This approach can reduce the computational overhead of repeatedly allocating and deallocating memory in the method of Santillana et al. (2010). We then determine locally which chemical regime to apply on basis of the ensemble of species classified as fast.

Construction of the chemical regimes can be done objectively by searching for a minimum in the computational cost of the chemical operator over the global domain. But some narrowing of the search is necessary. For the 228-species mechanism in GEOS-Chem, there are in principle $2^{228}$-1 possible combinations of species that would form mechanism subsets. The vast majority of those combinations make no chemical sense, but diagnosing this objectively would be computationally unfeasible. Instead, we start by splitting the mechanism species into $N$ different blocks based on similarity of chemical behavior. Then we classify a block as fast if at least one species in the block is fast, and slow if all species in the block are slow. The chemical regime is then defined as the assemblage of ,fast blocks, for which all species are treated as fast.

The partitioning of species into blocks can be optimized by minimizing globally the number of fast species (and hence the computation cost) for a given threshold $\delta$. We use for this purpose a training dataset from a GEOS-Chem simulation for 2013, consisting of the global ensemble of tropospheric and stratospheric gridboxes for the first 10 days of February, May, August, and November sampled every 6 hours. For each gridbox $j$, we diagnose each individual species $i$ as fast or slow following Section 3.1. We then diagnose the blocks as fast or slow with the indicator $y_{i,j}$ = 1 if the block is fast (at least one species in the block is fast) or $y_{i,j}$ = 0 if the block is slow (all species in the block are slow). The fraction $Z_1$ of all species that needs to be treated as fast over the testing domain is then given by

$$Z_1 = \frac{1}{\Omega}\sum_i \sum_j y_{i,j} \quad (6)$$

where $\Omega$ is the total number of gridboxes multiplied by the total number of species (228 in our case). We seek the partitioning of species into blocks that will minimize $Z_1$, and we use for that purpose the simulated annealing algorithm (Kirkpatrick et al., 1983). Starting from an arbitrary partitioning of the 228 species into $N$ blocks, and at each iteration of the algorithm, we randomly move one species from one block to another. If $Z_1$ decreases, this transition is accepted; if not, the transition is accepted with a probability controlled by a parameter named temperature that decreases gradually as the algorithm proceeds. Among the $N$ blocks, 3 are allocated to the reactive inorganic halogen species, and $N$-3 are allocated to the other species. This forced separation of the reactive inorganic halogen species is because the corresponding blocks are imposed to be fast above 10 km altitude (see Section 3.1).

The chemical regimes are then defined as different assemblages of blocks. This yields $2^N$ - 1 possible chemical regimes. Individual gridboxes in the model domain may correspond to any of these $2^N – 1$ regimes depending on which blocks are classified as fast or slow. We need to limit the number of regimes to a much smaller number $M$ of most useful regimes in order to keep the compilation of the code manageable, and as we will see the bulk of conditions in the model domain can effectively be represented by just a few regimes. Gridboxes that do not correspond to any of the $M$ regimes need to be matched to one of the $M$ regimes by moving some blocks from slow to fast, which will change the values of the corresponding indicators $y_{i,j}$ from 0 to 1. We refer to $y^*_{i,j}$ as the indicators adjusted by these changes. Thus the fraction $Z_2$ of species that needs to be treated as fast over the global domain is given by:

$$Z_2 = \frac{1}{\Omega}\left(\sum_{D_2}\sum_j y_{i,j} + \sum_{D_2}\sum_j y^*_{i,j}\right) \qquad (7)$$

where $D_1$ are the gridboxes that can be represented by the top $M$ chemical regimes, and $D_2$ are the gridboxes that are represented by other regimes and must be matched to the top $M$ regimes.

We tested a range of values from 3 to 20 for the number $N$ of blocks. In this testing we used a threshold $\delta$ = 100 molecules $cm^{-3}$ $s^{-1}$ to partition fast and slow species, following Santillana et al. (2010), and a number $M$ = 20 of chemical regimes (see next paragraph for choice of $M$). Figure S1 shows the fraction of fast species in the global domain ($Z_2$) as a function of $N$. If $N$ is low such that blocks are large, there is more likelihood that a species in a given block will be fast causing all species in the block to be treated as fast. If $N$ is high, more blocks will need to be moved from slow to fast in order to match the limited number $M$ of chemical regimes. We thus find an optimal value $N$ = 12 at which only 40% of the species need to be treated as fast. Table 1 lists selected representative species of these 12 blocks and a full listing is in Fig. S2. The blocks generally group species with coherent chemical behavior but there are some unexpected groupings (such as sulfur species and

peroxyacetylnitrate) and separations (such as $HO_2$ and $H_2O_2$) resulting from the optimization. Oxidants such as OH, $O_3$, and $NO_2$ are important under all circumstances so block 9 is fast in all gridboxes. Nonmethane VOCs species often have low concentrations outside of the continental boundary layer, and very low concentrations in the stratosphere, so the dominant VOC blocks 1-7 are fast in fewer than 40% of gridboxes.

We then tested different numbers of chemical regimes ($M$) from 3 to 40 for combining the $N = 12$ blocks, and again selected the regimes to minimize the global fraction $Z_2$ of species to be included in the implicit solver. $Z_2$ decreases from 65% to 40% as $M$ increases from 3 to 20 and flattens at higher values of $M$ (Fig. 1a). This is because 88% of the gridboxes can be represented by 20 chemical regimes (Fig. 1b). A larger number of blocks ($N > 12$) would extend the improvement to higher values of $M$, but the size of $M$ is also limited by considerations of code manageability and compilation speed. We use 20 chemical regimes in what follows.

Table 2 shows the composition of the 20 chemical regimes as defined by the blocks of Table 1. For 72% of the gridboxes, we only need to solve for fewer than 50% of the species as fast. Only 3.6% of gridboxes need to use the full chemistry mechanism, as defined by the 20$^{th}$ regime.

Figure 2 further shows the distribution of these 20 chemical regimes globally and for different altitudes, and the corresponding percentage of fast species that needs to be included in the chemical solver. In continental surface air where VOC emissions are concentrated, we find that over 80% of species generally need to be included. This percentage is reduced to 20-60% over the ocean and < 20% over Antarctica. At 5 km altitude, we find a distinct boundary between the daytime and nighttime hemisphere; the daytime chemistry is more active, and the percentage of fast species is higher in the daytime (40-60%) than at night (10%-30%). At 15 km altitude the extratropics are in the stratosphere, where non-methane VOC chemistry is largely absent, but the model still needs to solve 30-40% species as fast because of the halogens. Deep convection over tropical continents delivers short-lived VOCs and their oxidation products to the upper troposphere, so that a large number of species needs to be treated as fast in the convective outflow where and when it occurs. The importance of deep convective outflow for global atmospheric chemistry has been pointed out in a number of studies (Prather and Jacob, 1997; Bechara et al., 2010; Schroeder et al., 2014), and emphasizes the advantage of reducing the mechanism on the fly rather than with pre-set geographic boundaries.

## 4 Error analysis

Here we quantify the errors in our adaptive reduced mechanism method by comparison with a standard GEOS-Chem simulation for the troposphere and stratosphere (version 12.0.0) including full chemistry (228 species). The comparison is conducted for a 1-month simulation to examine the sensitivity to the rate threshold $\delta$, and for a 2-year simulation to evaluate

the stability of the method. In both cases, we use the Relative Root Mean Square (RRMS) metric as given by Sandu et al. (1997) to characterize the error:

$$RRMS_i = \sqrt{\frac{1}{Q_i}\sum_{j=1}^{Q_i}\left(\frac{n_{i,j}^{\text{reduced}} - n_{i,j}^{\text{full}}}{n_{i,j}^{\text{full}}}\right)^2} \quad (8)$$

Where $n_{i,j}^{\text{reduced}}$ and $n_{i,j}^{\text{full}}$ are the concentrations for species $i$ and gridbox $j$ in the reduced and full chemical mechanisms, and the sum is over the the gridboxes where $n_{i,j}^{\text{full}}$ is greater than a threshold $a$, and $Q_i$ is the number of such gridboxes. Here we use $a = 1\times10^6$ molecules cm$^{-3}$ as in Eller et al. (2009) and Santillana et al. (2010).

A critical parameter to select in the algorithm is the rate threshold $\delta$ separating fast and slow species on the basis of their production and loss rates. A high threshold decreases the number of fast species and hence speeds up the computation but at the expense of accuracy. We tested different rate thresholds ranging from 10 to 5000 molecules cm$^{-3}$ s$^{-1}$ in a 1-month GEOS-Chem simulation starting on August 1 2013. Figure 3 shows the median RRMS error for all species on September 1 and the increased computational performance for different rate thresholds $\delta$. The best range for $\delta$ is between 100 and 1000 molecules cm$^{-3}$ s$^{-1}$, where the median RRMS error is below 1% and the improvement in computational performance is in the 30-40% range.

Figure S3 further shows the distribution of RRMS errors over all species for different rate thresholds $\delta$. The 90$^{th}$ percentile RRMS error stays below 5% if $\delta \leq 1000$ molecules cm$^{-3}$ s$^{-1}$ but exceeds 10% for $\delta = 5000$ molecules cm$^{-3}$ s$^{-1}$. The 99$^{th}$ percentile RRMS error is less than 20% for $\delta \leq 1000$ molecules cm$^{-3}$ s$^{-1}$ but rises to 80% for $\delta = 5000$ molecules cm$^{-3}$ s$^{-1}$. The largest errors are usually from the tropospheric halogen species (Fig. S4). When near the day-night terminator, the sharp transition of production and loss rates are difficult to be approximated using the first-order explicit equations, resulting in high relative errors.

Figure 4 shows the time evolution over two years of simulation of the median RRMS error for all species and also for the selected species OH, ozone, sulfate, and $NO_2$. Here we used a rate threshold $\delta$ of 500 molecules cm$^{-3}$ s$^{-1}$. The median RRMS for all species is below 0.6%. It is below 0.3% for OH, ozone, and sulfate, and below 2% for $NO_2$. Results for thresholds of 100 and 1000 molecules cm$^{-3}$ s$^{-1}$ are in Fig. S5. We find that the error remains stable over time. Figure 5 displays the spatial distribution of the relative error on the last day of the 2-year simulation. The relative errors are below 0.5% everywhere for $O_3$, OH, and sulfate. The error for $NO_2$ reaches 1-10% at high latitudes, but this is still well within other systematic sources of errors in estimating $NO_2$ concentrations (Silvern et al., 2019).

## 5 Conclusions

We have presented an adaptive method to speed up the temporal integration of chemical mechanisms in global atmospheric

chemistry models. This integration ("chemical operator") involves the implicit solution of a stiff coupled system of ordinary differential equations (ODEs) representing the kinetic evolution of individual species in the mechanism. With typical mechanisms including over 100 coupled species, this chemical integration is the principal computational bottleneck in atmospheric chemistry models and hinders the adoption of detailed atmospheric chemistry in Earth system models.

Our method takes advantage of the fact that different regions of the atmosphere need different levels of detail in the chemical mechanism, and that greatly reduced mechanisms can be used in most of the atmosphere. We do this reduction locally and on the fly by choosing from a portfolio of pre-selected reduced chemical mechanisms (chemical regimes) on the basis of species production and loss rates, distinguishing between "fast" species that need to be in the coupled mechanism and "slow" species that can be solved explicitly. Our method has five fundamental advantages over other methods proposed to speed up the chemical computation: (1) It does not sacrifice the complexity of the chemical mechanism where it is needed, while greatly simplifying it over much of the world where it is not. (2) It conserves all of the meaningful diagnostic information of the chemical system, such as production and loss rates of species and families, and individual reaction rates. (3) It can be tailored to achieve the level of simplification that one wishes. (4) It is robust against small mechanistic changes, as these may not alter the choice of chemical regimes or may be accommodated by minor tweaking of the regimes. (5) It is robust against increases in model resolution, where source gridboxes (e.g., urban areas) will simply default to the full mechanism.

We applied the method to the GEOS-Chem global 3-D model for oxidant-aerosol chemistry in the troposphere and stratosphere. The full chemical mechanism in GEOS-Chem has 228 coupled species. We developed an objective numerical method to pre-select the reduced chemical regimes on the basis of time slices of full-mechanism model results. We showed that 20 regimes could cover efficiently the range of atmospheric conditions encountered in the model. We then pick appropriate regimes for the chemical operator on the fly by comparing the local production and loss rates of individual model species to a threshold $\delta$. Values of $\delta$ in the range 100-1000 molecules cm$^{-3}$ maintain an accuracy better than 1% relative to a model simulation with the full mechanism and decrease the computational cost of the chemical solver by 32-41%. Comparison testing with a 2-year global GEOS-Chem simulation for the troposphere and stratosphere including the full mechanism shows errors of less than 1% for critical species and no significant error growth over the two years.

Several improvements could be made to our method to further reduce computational costs. (1) The blocks of species used to construct the reduced chemical mechanisms are optimized to minimize the number of fast species but are not always chemically logical, which could be improved by applying prior constraints to the optimization. (2) Optimization in the definition of the reduced mechanisms could take into account not only the number of species but also their lifetimes that affect the stiffness of the system. (3) Separation between fast and slow species could take into account species lifetimes, because species with long lifetimes but high loss rates (such as methane or CO) can be solved explicitly. (4) Mass conservation in the explicit solution could be enforced to enable more species (in particular stratospheric halogens) to be treated explicitly when they play little role in the coupled system. (5) Besides removing the slow species from the implicit

chemical operator, we could also remove unimportant reactions, which would reduce the cost in updating the production/loss rates and the Jacobian matrix. These improvements will be the target of future work.

**Code availability**. The standard GEOS-Chem code is available through https://doi.org/10.5281/zenodo.1343547. The updates for the adaptive mechanism can be found at https://doi.org/10.7910/DVN/IM5TM4.

**Data availability**. All datasets used in this study are publically accessible at https://doi.org/10.7910/DVN/IM5TM4.

**Author contribution.** L. Shen and D. Jacob designed the experiments and L. Shen carried them out. L. Shen and D. Jacob prepared the manuscript with contributions from all co-authors.

**Competing Interests**. The authors declare that they have no conflict of interest.

**Acknowledgments.** This work was funded by the NASA Modeling and Analysis Program (MAP)

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

**Table 1.** Partitioning of GEOS-Chem chemical species into $N = 12$ blocks[a]. Results are for August 1 2013.

| Block | Type of species[b] | Number of species[c] | % gridboxes where fast[d] |
|---|---|---|---|
| 1 | Aromatics | 21 | 33.4 |
| 2 | Organic nitrates | 7 | 39.3 |
| 3 | Isoprene, terpenes | 30 | 13.9 |
| 4 | Alkanes, alkenes, acetone | 12 | 41.4 |
| 5 | Higher alkanes, methyl ethyl ketone | 14 | 36.5 |
| 6 | Halocarbons | 55 | 10.2 |
| 7 | Secondary organic aerosol | 25 | 15.6 |
| 8 | Sulfur, peroxyacetylnitrate | 14 | 95.9 |
| 9 | Oxidants | 12 | 100.0 |
| 10 | Iodine reservoirs | 13 | 69.5 |
| 11 | Bromine and chlorine inorganic species | 11 | 99.9 |
| 12 | Bromine and iodine radicals | 13 | 85.0 |

[a]The full GEOS-Chem mechanism has 228 species. The blocks defined here are combined in the algorithm to generate $M$=20 different chemical regimes as subsets of the full mechanism.

[b]Qualitative descriptor of the most important species in the block.

[c]Figure S2 lists all species in each block.

[d]Global percentage of GEOS-Chem model gridboxes in the troposphere and stratosphere where the block is treated as fast.

**Table 2**. Composition and frequency of the 20 chemical regimes in the adaptive algorithm[a]. Results are for August 1 2013.

| | Block[b] | | | | | | | | | | | | % fast species [b] | % gridboxes |
|---|---|---|---|---|---|---|---|---|---|---|---|---|---|---|
| Regime # | 1 | 2 | 3 | 4 | 5 | 6 | 7 | 8 | 9 | 10 | 11 | 12 | | |
| 1 | 0 | 0 | 0 | 0 | 0 | 0 | 0 | 0 | 1 | 0 | 0 | 0 | 5.6 | 0.1 |
| 2 | 0 | 0 | 0 | 0 | 0 | 0 | 0 | 0 | 1 | 0 | 1 | 0 | 10.3 | 3.9 |
| 3 | 0 | 0 | 0 | 0 | 0 | 0 | 0 | 0 | 1 | 0 | 1 | 1 | 15.8 | 0.1 |
| 4 | 0 | 0 | 0 | 0 | 0 | 0 | 0 | 1 | 1 | 0 | 1 | 0 | 18.4 | 5.4 |
| 5 | 0 | 1 | 0 | 0 | 0 | 0 | 0 | 1 | 1 | 0 | 1 | 0 | 21.4 | 2.2 |
| 6 | 0 | 0 | 0 | 0 | 0 | 0 | 0 | 1 | 1 | 0 | 1 | 1 | 23.9 | 0.5 |
| 7 | 0 | 1 | 0 | 0 | 0 | 0 | 0 | 1 | 1 | 0 | 1 | 1 | 26.9 | 0.2 |
| 8 | 0 | 1 | 0 | 1 | 0 | 0 | 0 | 1 | 1 | 0 | 1 | 0 | 26.9 | 1.1 |
| 9 | 0 | 0 | 0 | 0 | 0 | 0 | 0 | 1 | 1 | 1 | 1 | 1 | 29.5 | 46.3 |
| 10 | 0 | 0 | 0 | 1 | 0 | 0 | 0 | 1 | 1 | 0 | 1 | 1 | 29.5 | 0.5 |
| 11 | 0 | 0 | 0 | 1 | 0 | 0 | 0 | 1 | 1 | 1 | 1 | 1 | 35.0 | 3.3 |
| 12 | 0 | 0 | 0 | 1 | 1 | 0 | 0 | 1 | 1 | 0 | 1 | 1 | 35.5 | 0.7 |
| 13 | 0 | 1 | 0 | 1 | 1 | 0 | 0 | 1 | 1 | 0 | 1 | 1 | 38.5 | 2.4 |
| 14 | 1 | 1 | 0 | 1 | 1 | 0 | 0 | 1 | 1 | 0 | 1 | 1 | 47.4 | 5.2 |
| 15 | 1 | 1 | 0 | 1 | 1 | 0 | 0 | 1 | 1 | 1 | 1 | 1 | 53.0 | 12.7 |
| 16 | 1 | 1 | 0 | 1 | 1 | 0 | 1 | 1 | 1 | 0 | 1 | 1 | 58.1 | 1.7 |
| 17 | 1 | 1 | 1 | 1 | 1 | 0 | 1 | 1 | 1 | 1 | 1 | 1 | 76.5 | 3.7 |
| 18 | 1 | 1 | 1 | 1 | 1 | 1 | 1 | 1 | 1 | 0 | 1 | 0 | 88.9 | 2.3 |
| 19 | 1 | 1 | 1 | 1 | 1 | 1 | 1 | 1 | 1 | 0 | 1 | 1 | 94.4 | 4.4 |
| 20 | 1 | 1 | 1 | 1 | 1 | 1 | 1 | 1 | 1 | 1 | 1 | 1 | 100.0 | 3.6 |

[a]The chemical regimes are defined by the ensemble of fast species that need to be treated as a coupled system with implicit solution in the chemical operator. The species are assembled into blocks as summarized in Table 1, and here we identify the blocks treated as fast in the chemical regime (1 ≡ fast, 0 ≡ slow).

[b]The composition of each block is defined in Table 1 and Figure S2.

[b]Percentage of the 228 species in the GEOS-Chem chemical mechanism treated as fast in the chemical regime.

[c]Global percentage of GEOS-Chem tropospheric and stratospheric gridboxes for which the chemical regime is selected.

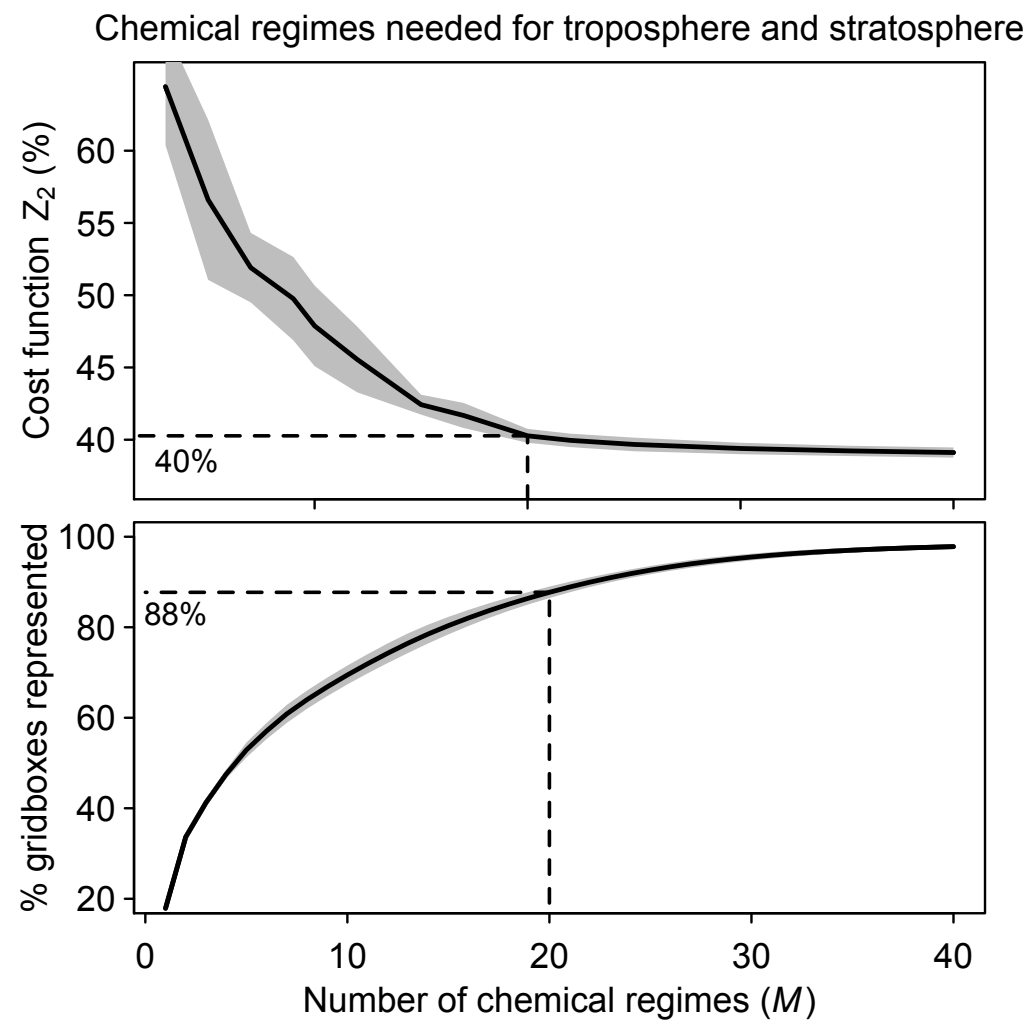

**Figure 1**. Speed-up of the chemical computation as a function of the number $M$ of chemical mechanism subsets (chemical regimes) used in the coupled implicit solver of the GEOS-Chem model for adaptive simulation of the troposphere and stratosphere. Top: Cost function $Z_2$ (global fraction of chemical species treated as fast) as a function of the number of chemical regimes. Bottom: Percentage of model gridboxes that can be represented by the $M$ chemical regimes without adjustment (see text). Dashed lines show the values for $M = 20$. Results are for the first 10 days of February, May, August, and November sampled every 6 hours (shaded area denotes one standard deviation).

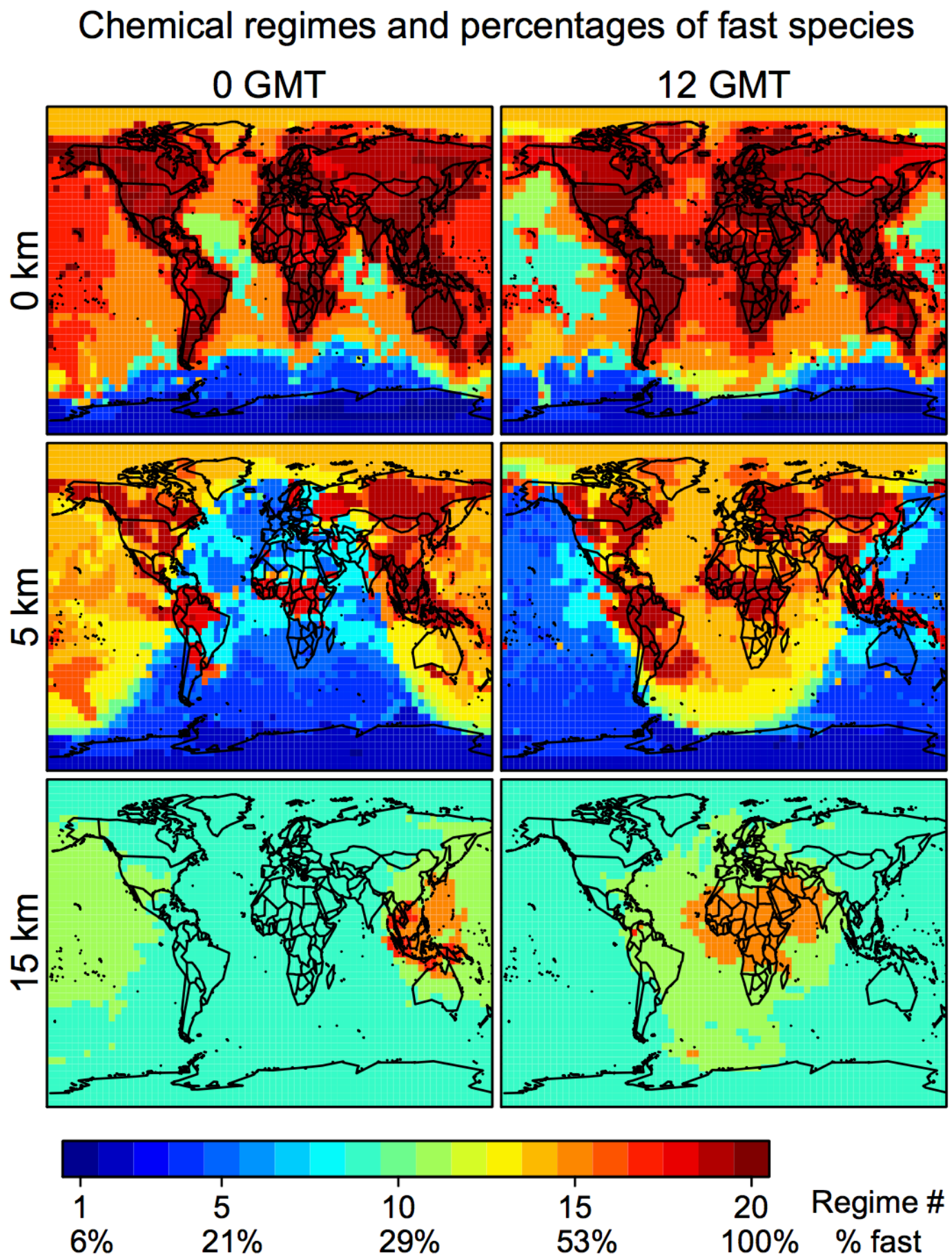

**Figure 2.** Chemical mechanism complexity needed in different regions of the atmosphere. The Figure identifies the chemical regime from Table 2 needed to simulate a given GEOS-Chem gridbox on August 1 2013 at 0 and 12 GMT. The percentage of species treated as fast (requiring coupled implicit solution) in that chemical regime is shown on the colorbar and more details are in Tables 1 and 2. Results are shown for different altitudes and using a threshold $\delta$ of 100 molecules $cm^{-3}$ $s^{-1}$.

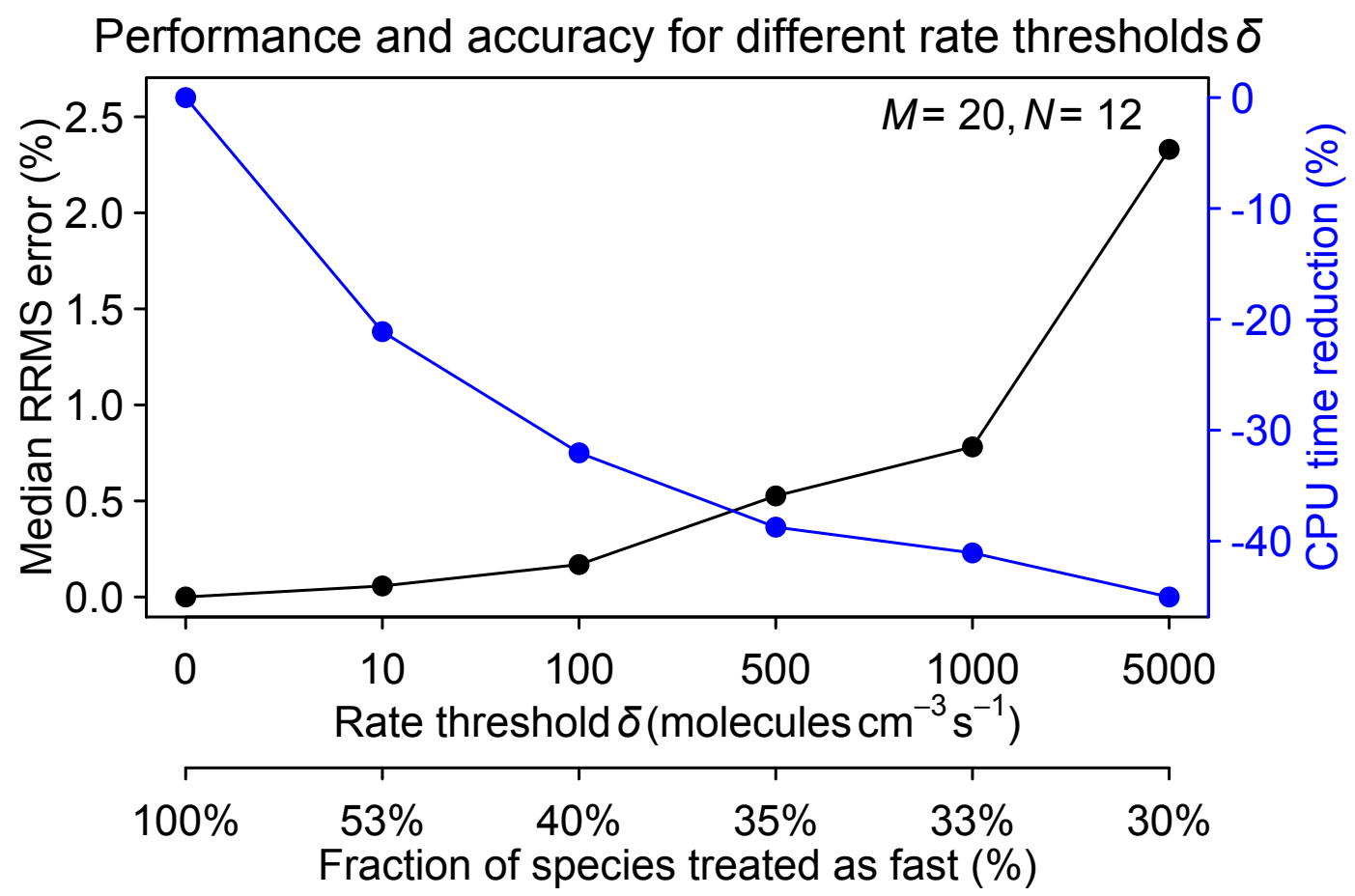

**Figure 3**. Performance and accuracy of the adaptive chemical mechanism reduction method for different rate thresholds δ (molecules $cm^{-3}$ $s^{-1}$) to separate fast and slow species. The performance is measured by the reduction in computing processor unit (CPU) time for the chemical operator, and the accuracy is measured by the median relative root mean square (RRMS) error for species concentrations relative to a global GEOS-Chem simulation for the troposphere and stratosphere using the full chemical mechanism. The second *x* axis gives the global fraction of species that need to be treated as fast depending on the value of δ. The number of blocks (*N*) is 12 and the number of chemical regimes (*M*) is 20.

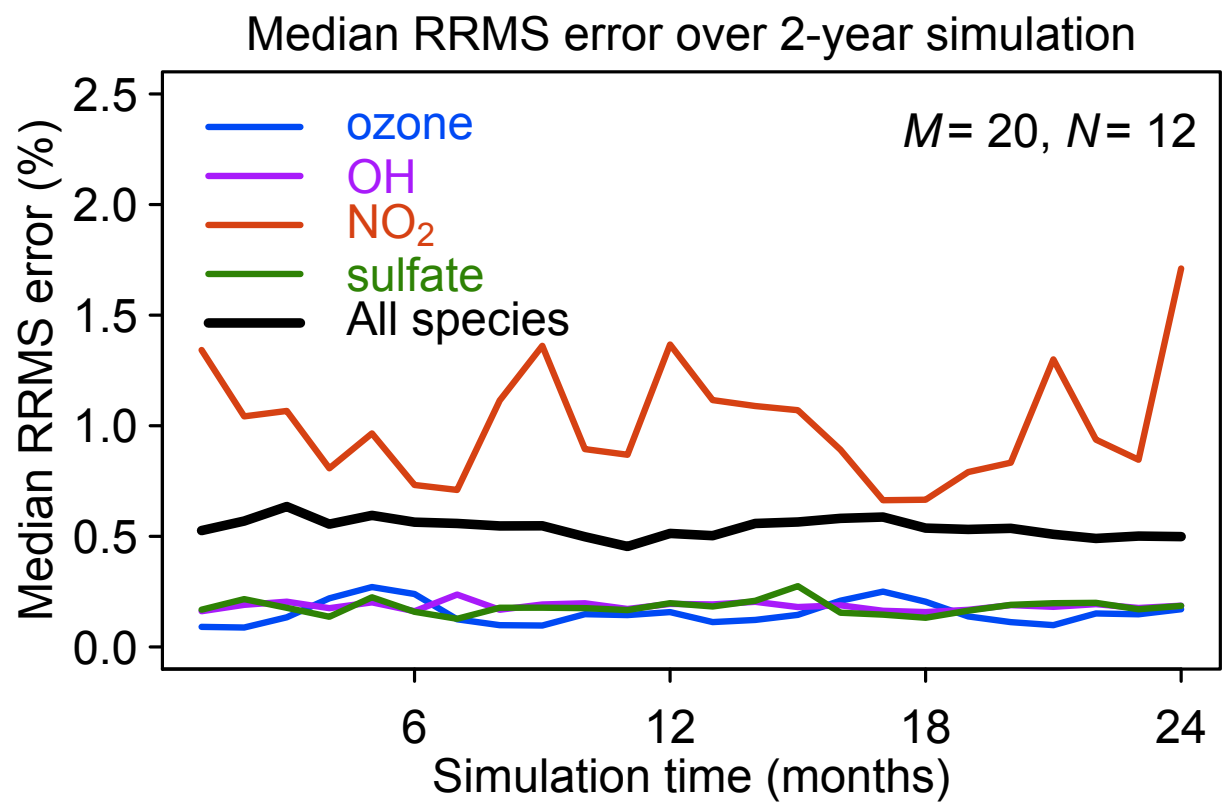

**Figure 4**. Accuracy of the adaptive reduced chemistry mechanism algorithm over a two-year GEOS-Chem simulation (see text). The accuracy is measured by the RRMS error relative to a simulation including the full chemical mechanism. A rate threshold $\delta$ = 500 molecules cm$^{-3}$ s$^{-1}$ is used to partition the fast and slow species in the reduced mechanism. Results are shown for the median RRMS across all species in the mechanism and more specifically for ozone, OH, $NO_2$, and sulfate.
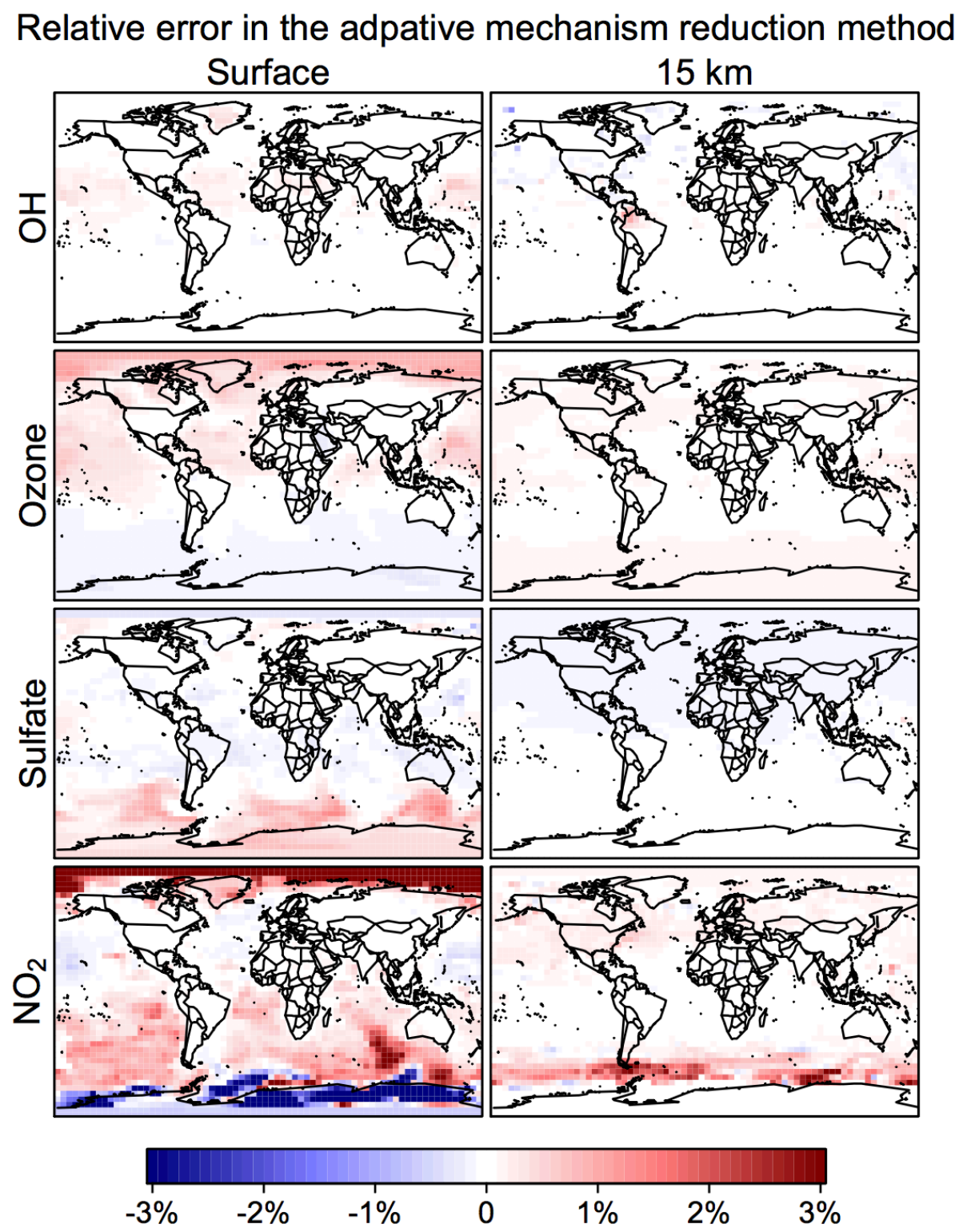

**Figure 5**. Relative error from the adaptive mechanism reduction method after two years of simulation in the GEOS-Chem global 3-D model for tropospheric-stratospheric chemistry. The figure shows relative differences of 24-h average OH, ozone, sulfate and $NO_2$ concentrations relative to the full-chemistry simulation on the last day of the two-year simulation (2013-2014). The relative error for surface $NO_2$ can be up to ±10% in polar regions. The calculation uses a rate threshold $\delta = 500$ molecules $cm^{-3}$ $s^{-1}$ to partition the species between fast and slow. The number of blocks ($N$) is 12 and the number of chemical regimes ($M$) is 20.