# Peer review of "An adaptive method for speeding up the numerical integration of chemical mechanisms in atmospheric chemistry models: application to GEOS-Chem version 12.0.0"

_Geoscientific Model Development, 2019_

## Referee Comment (RC1) · Anonymous Referee #1 · 11 Nov 2019

The integration of the chemical differential equations in a chemistry transport model is a significant computational burden. Both climate and air quality forecasting models spend a significant fraction of their computation marching forwards through time, solving these differential equations. Developments in speeding up this code have been not been forthcoming over the last decades and it has only really been the availability of more, faster CPU cores that has allowed us to run increasingly complex chemistry. Algorithmic developments to speed up the integration of this chemistry are to be welcomed.

[Figure]
Interactive
comment

This paper outlines a method for running different subsets of the differential equations in different geographical regions, so speeding up the solutions. On the one hand, this is a fairly obvious thing to do. We don't need to be integrating all of the chemistry of isoprene over the middle of the Pacific in the same way as we do over the middle of the Amazon. However, attempting this adaptive chemistry has not been readily taken up by the community. This paper attempts a first realistic attempt at such an implementation.

In general, the paper describes a novel new technique with a potentially sound methodology which could be extremely useful to the atmospheric chemistry transport modelling community. At this point though, I am not convinced that the algorithm is working as intended, or if it is, the explanations given in the text are satisfactory for the reader to understand what is going on. If either of these outcomes can be corrected the paper should be published.

1) My major concern about the methodology is the split of species into the different blocks. The methodology for doing this is explained but there is little interpretation of the results. In many cases, the blocks seem to have lumped together some fairly random sets of species and this appears to have been hidden away in the SI.

Block 1: "Aromatics" (Benzene and Toluene but the chemically almost identical XYLE is in Block 7?) but it also contains $CH_2I_2$. What is the advantage of using the integrator for these together? Does this mean that the aromatic chemistry is also being solved with the integrator in the middle of the oceans where the $CH_2I_2$ is important? Block 2: "Organic nitrates" This contains some organic nitrates and the N atom. These organic compounds are only really at appreciable concentrations at the surface whereas the N atoms are only really applicable in the upper stratosphere? Block 3: "Isoprene" This seems to contain some isoprene chemistry but also HFCS which seems suprising. Block 6: "Halocarbons" seems to be again mainly isoprene species to me, but it also contains HFC and CFC species which would only be important in the stratosphere.

Is there an explanation for this? Are the blocks in the SI correct? I might be missing an

important concept here. But it has not been explained. To me, this feels like an error has occurred somewhere either in the species list given in the SI or in the algorithm.

Given the random nature of the annealing algorithm, is this set of species blocks always the same one? What degree of variation is present when running this algorithm multiple times?

There needs to be more work done to explain why these blocks are the best ones to use given the variation seen in the species type in each block. I realise this is a result of the optimization algorithm but the situation at the moment appears to be that an algorithm has told us that this is the result and we are going to leave it at that. To my mind, the species within these blocks do not appear to have the properties you would expect given what is trying to be achieved. I'm happy to be convinced otherwise but the text at the moment does not achieve that and it is not possible therefore to be confident that the algorithms are working appropriately.

2) The Supplementary information figures should be contained within the main text. The paper is fairly short and some of the figures are central to understanding the methodology. Figure S2 could be removed by putting the species list into Table 1. The other figures are small enough to be included in the main body without overwhelming the reader.

3) I found the structure of the beginning of the paper a bit confused. We have an Introduction; a section on the chemical operator which includes a very brief description of the GEOS-Chem model and the KPP system. We then have a section on the algorithm being described. The material about the chemical operator should be moved into the introduction as this basically supports the introductory text about the chemical integration. The model description should go into a separate section. 4) Page 1 Line 30. The number of reactions thought in play in atmospheric chemistry is significantly more than the "hundreds" described in the text. The MCM has 10s of thousands and mechanisms produced by GECKO-A produces millions. Hundreds are used in the sim-

plified mechanisms for atmospheric chemistry transport models. The text should be clarified here.

5) Page 2 lines 55. There are now some other approaches to speed up the chemical integration using "Machine Learning" approaches they could be cited here.

6) Page 6 line 155 the reference to Santillana is 210 rather than 2010.

7) Page 6 line 166. Do the authors mean 'fast blocks' rather than ,fast blocks,?

8) Figure 1. When asking the reader to "see text" can this be more specific? What does the shaded area represent? The SD between what, the monthly values?

9) Figure 2. Although using this approach does provide some information it would be useful to split the dataset in another way. Could there be a figure which shows a map of the world indicating whether each block is switched on at that location. This need only be done at the surface for 0 GMT and 12 GMT but it would give some confidence that the approach is working. isoprene block should only be on over continental regions etc. It is very hard to get this level of information from the figures as presented. Why is the value of delta of 100 used in this figure and 500 used in other figures?

10) Figure 4. Can the figure caption give more information here? What actually is being compared? Is this the RRMS in the monthly mean fields, or in the hourly values averaged to a monthly mean? Is this all of the species in the Jacobian?

11) It's not obvious that the code for the annealing algorithm is included in the repository. I've had a look but can't find it.

12) Conclusions. a. It would be useful to discuss whether this algorithm could be used within the adjoint framework for data assimilations, inversion studies? b. The authors discuss the suitability of this approach to minor mechanistic changes. However, if the algorithm is to be useful it needs to be sustainable within the software lifecycle of the chemistry transport model. Could this be spelt out in more detail? Presumably, if a new species was added the training algorithm (which species into which block and

how many blocks etc) would need to the re-run with new data, but a small change in species lifetime would not lead to a re-running. It would be useful to have the conditions which are required for the training to be updated to be described.

———————————————

---

## Referee Comment (RC2) · Anonymous Referee #2 · 2 Dec 2019

Shen et al describes the implementation of a method of reducing the computational complexity of solving a chemical mechanism within GEOS-chem. The paper is interesting, although further revisions are required before it can be considered for publication.

**Major Comments**

I found the discussion in Section 3.2 very hard to follow, specifically how blocks are grouped into regimes and then the subsequent changing of blocks from slow to fast if

a gridbox does not correspond to any of the regimes. The sentence in question is

"Gridboxes that do not correspond to any of the M regimes need to be matched to one of the M regimes by moving some blocks from slow to fast, which will change the values of the corresponding indicators $y_{i,j}$ from 0 to 1."

Could the authors explain just how the mapping of species to blocks to regimes to these re-matched regimes is done? A diagram or pseudocode would be useful here. This crucial step is not explained well, and I'm not sure if this step is done online or not. How is the regime determined during a model run, and how is it ensured that the regimes are correctly matched (and what happens when they do not match)? This information is required to adequately understand the method presented.

I would be interested to know how robust the particular organisation determined from the Simulated Annealing algorithm is. Were multiple simulated annealing simulations performed? Was the rate of reduction of "temperature" changed to see if this affected the results? As with any global optimisation technique it is possible to get stuck in local minima, and a single run-through this algorithm will not be sufficient to determine whether the true minima has been found.

While not essential for this manuscript, I would be interested to know if this classification has any load-balancing implications. I can imagine that for codes with MPI parallelisation across many nodes of a HPC, this method will increase the imbalance between different MPI tasks (while still decreasing the overall run-time). This could then lead to further speed improvements if the load-balancing is improved.

Most of the discussion and plots presented use a $\delta$ of 100 molecules cm$^{-3}$ s$^{-1}$ (or a range is presented), except when $\delta$ = 500 is used for Figures 4 (the equivalent plot for $\delta$ = 100 is Figure S5) and 5 and the discussion surrounding the 2-year runs in Section 4. Given that the $\delta$ = 100 results seem noticeably better, why were the $\delta$ = 500 presented in the main text? Are there equivalent plots (especially the Figure 5 equivalents) for the other values of $\delta$ used (100,1000)?

Given the errors associated with halogen species presented in Figure S4, would there be a large drop in performance if these species were always treated as fast?

**Minor Corrections**

Page 6, Line 165: ",fast blocks,"

Page 7, Equation 7: There is no $D_1$, both $\Sigma$ are labelled with $D_2$

Page 16, Figure 1: The X-axes for the panels are slightly off-set. This can be clearly seen in the downward dotted lines.
* * *

---

## Author Response (AR1)

**Response to referee comments on "An adaptive method for speeding up the numerical integration of chemical mechanisms in atmospheric chemistry models: application to GEOS-Chem version 12.0.0"**

We thank the referees for their careful reading of the manuscript and the valuable comments. This document is organized as follows: the Referee's comments are in *italic*, our responses are in plain text, and all the revisions in the manuscript are shown in blue. **Boldface blue text** denotes text written in direct response to the Referee's comments. The line numbers in this document refer to the updated manuscript.

**Reviewer 1**

*The integration of the chemical differential equations in a chemistry transport model is a significant computational burden. Both climate and air quality forecasting models spend a significant fraction of their computation marching forwards through time, solving these differential equations. Developments in speeding up this code have been not been forthcoming over the last decades and it has only really been the availability of more, faster CPU cores that has allowed us to run increasingly complex chemistry. Algorithmic developments to speed up the integration of this chemistry are to be welcomed.*

*This paper outlines a method for running different subsets of the differential equations in different geographical regions, so speeding up the solutions. On the one hand, this is a fairly obvious thing to do. We don't need to be integrating all of the chemistry of isoprene over the middle of the Pacific in the same way as we do over the middle of the Amazon. However, attempting this adaptive chemistry has not been readily taken up by the community. This paper attempts a first realistic attempt at such an implementation.*

*In general, the paper describes a novel new technique with a potentially sound methodology which could be extremely useful to the atmospheric chemistry transport modelling community. At this point though, I am not convinced that the algorithm is working as intended, or if it is, the explanations given in the text are satisfactory for the reader to understand what is going on. If either of these outcomes can be corrected the paper should be published.*

**Response**. Thanks for raising these good points. This feedback has significantly improved the manuscript.

*1) My major concern about the methodology is the split of species into the different blocks. The methodology for doing this is explained but there is little interpretation of the results. In many cases, the blocks seem to have lumped together some fairly random sets of species and this appears to have been hidden away in the SI.*

**Response**. Thanks, we have added more interpretation of the results and one independent paragraph to discuss the shortcoming of our present method. The blocks of species are constructed by minimizing the number of fast species, so it cannot guarantee the groups to be always chemically logical. We have a follow-up project to fix this issue by introducing a regularization term that defines the species' distances as learned from their reactant-product relationships. The preliminary result of the revised method is more chemically logical, but it still has some unresolved issues so we are not able to present it in this work. In this manuscript, we will make it clear that our present method can only generally group species with coherent chemical behaviors but there are unexpected groups.

*Block 1: "Aromatics" (Benzene and Toluene but the chemically almost identical XYLE is in Block 7?) but it also contains CH2I2. What is the advantage of using the integrator for these together? Does this mean that the aromatic chemistry is also being solved with the integrator in the middle of the oceans where the CH2I2 is important? Block 2: "Organic nitrates" This contains some organic nitrates and the N atom. These organic compounds are only really at appreciable concentrations at the surface whereas the N atoms are only really applicable in the upper stratosphere? Block 3: "Isoprene" This seems to contain some isoprene chemistry but also HFCS which seems suprising. Block 6: "Halocarbons" seems to be again mainly isoprene species to me, but it also contains HFC and CFC species which would only be important in the stratosphere.*

*Is there an explanation for this? Are the blocks in the SI correct? I might be missing an important concept here. But it has not been explained. To me, this feels like an error has occurred somewhere either in the species list given in the SI or in the algorithm.*

**Response.** Our present method cannot perfectly separate these species by their chemical properties. We now discuss this shortcoming in one independent paragraph. Again, we will address this issue in a follow-up paper.

Line 216. This algorithm still has shortcomings. There are some unexpected groupings (such as sulfur species and peroxyacetylnitrate) and separations (such as $HO_2$ and $H_2O_2$). The blocks are constructed by minimizing the number of fast species in the optimization, so species tend to be in the same block as long as they are fast or slow simultaneously. For example, isoprene products and CFCs are both slow in the stratosphere and clean regions, so they may be assigned into the same group (e.g., block 6). In addition, there are still noticeable changes of species groups if we run the simulated annealing algorithm with different initializations and choices of the temperature parameter, even though the optimized blocks can generally separate the oxidants, anthropogenic VOCs, and biogenic VOCs (Table S1). Here we chose the set of groupings that minimized the cost function for a number of realizations of the algorithm. These two shortcomings may be addressed by introducing regularization terms in the cost function to enforce known species relationships, which will implement this in a follow-up study.

*Given the random nature of the annealing algorithm, is this set of species blocks always the same one? What degree of variation is present when running this algorithm multiple times?*
**Response.** Thanks for raising this good point. The set of species blocks are not always exactly the same due to the random processes in simulated annealing. But in general, they can separate the oxidants, anthropogenic VOCs, and biogenic VOCs. Now we say.
Line 220. In addition, there are still noticeable changes of species groups if we run the simulated annealing algorithm with different initializations and choices of the temperature parameter, even though the optimized blocks can generally separate the oxidants, anthropogenic VOCs, and biogenic VOCs (Table S1). Here we chose the set of groupings that minimized the cost function for a number of realizations of the algorithm. These two shortcomings may be addressed by introducing regularization terms in the cost function to enforce known species relationships, which will implement this in a follow-up study.

*There needs to be more work done to explain why these blocks are the best ones to use given the variation seen in the species type in each block. I realize this is a result of the optimization algorithm but the situation at the moment appears to be that an algorithm has told us that this is the result and we are going to leave it at that. To my mind, the species within these blocks do not appear to have the properties you would expect given what is trying to be achieved. I'm happy to be convinced otherwise but the text at the moment does not achieve that and it is not possible therefore to be confident that the algorithms are working appropriately.*
**Response.** Thanks for pointing this out. This is a shortcoming of our present method, and we will fix it in the future study.
Line 223. These two shortcomings may be addressed by introducing regularization terms in the cost function to enforce known species relationships, which will implement this in a follow-up study.

*2) The Supplementary information figures should be contained within the main text. The paper is fairly short and some of the figures are central to understanding the methodology. Figure S2 could be removed by putting the species list into Table 1. The other figures are small enough to be included in the main body without overwhelming the reader*
**Response**. Thanks. We have moved two figures back to the main text. The information of Figure S2 has been included into Table 1. We also have 7 new figures (Figure S1-5, S8-9) in the supplement to better support our discussion.

*3) I found the structure of the beginning of the paper a bit confused. We have an Introduction; a section on the chemical operator which includes a very brief description of the GEOS-Chem model and the KPP system. We then have a section on the algorithm being described. The material about the chemical operator should be moved into the introduction as this basically supports the introductory text about the chemical integration. The model description should go into a separate section.*

**Response**. Thanks for pointing this out. Since the introduction is already very long, so we decide to remove the first two paragraphs about the chemical operator. And we have also changed the section name to 'Model description'.

*4) Page 1 Line 30. The number of reactions thought in play in atmospheric chemistry is significantly more than the "hundreds" described in the text. The MCM has 10s of thousands and mechanisms produced by GECKO-A produces millions. Hundreds are used in the simplified mechanisms for atmospheric chemistry transport models. The text should be clarified here.*

**Response**. Now we say this.

Line 30. The complete Master Chemistry Mechanism (MCM, version 3.3, http://mcm.leeds.ac.uk/MCMv3.3.1/) consists of 5,832 species and 16,701 reactions. Atmospheric chemistry models use greatly simplified mechanisms, which still include hundreds of species coupled through production and loss pathways and with lifetimes ranging from less than a second to many years.

*5) Page 2 lines 55. There are now some other approaches to speed up the chemical integration using "Machine Learning" approaches they could be cited here.*

**Response**. Now we say.

Line 62. Machine learning algorithms have been developed to replace the role of the conventional chemical solver; but these methods have only been applied to simple scenarios and are subject to error growth as simulation time progresses (Keller and Evans, 2019).

*6) Page 6 line 155 the reference to Santillana is 210 rather than 2010.*

**Response**. Fixed, thanks.

*7) Page 6 line 166. Do the authors mean 'fast blocks' rather than ,fast blocks,?*

**Response**. It is a typo. Now fixed, thanks.

*8) Figure 1. When asking the reader to "see text" can this be more specific? What does the shaded area represent? The SD between what, the monthly values?*

**Response**. Now we say

Line 446. See Equation 5 and related text.

Line 447. For both panels, results are for the first 10 days of February, May, August, and November sampled every 6 hours (shaded area denotes one standard deviation of results sampled every 6 hours).

9) Figure 2. Although using this approach does provide some information it would be useful to split the dataset in another way. Could there be a figure which shows a map of the world indicating whether each block is switched on at that location. This need only be done at the surface for 0 GMT and 12 GMT but it would give some confidence that the approach is working. isoprene block should only be on over continental regions etc. It is very hard to get this level of information from the figures as presented. Why is the value of delta of 100 used in this figure and 500 used in other figures?

**Response**. We have four supplementary figures to display the regions where anthropogenic and biogenic VOCs are treated as fast.

Line 213. Anthropogenic VOC species (blocks 4 and 5) are found to be fast in boundary layer and daytime mid-troposphere (Figure S2-3). Biogenic VOC species have shorter lifetimes, so they are found to be fast only in lower and middle troposphere over the land (Figure S4-5).

Now we show the results for rate thresholds $\delta$ of 100, 500 and 1000 molecules cm$^{-3}$ s$^{-1}$. Figure 5 is also updated.

Line 262. The best range for $\delta$ is between 100 and 1000 molecules cm$^{-3}$ s$^{-1}$, where the median RRMS error is below 1% and the improvement in computational performance is in the 30-40% range.

Line 271. Figure 5 shows the time evolution over two years of simulation of the median RRMS error for all species and also for the selected species OH, ozone, sulfate, and NO$_2$. The median RRMS for all species is 0.2%, 0.5%, and 0.8% for rate thresholds $\delta$ of 100, 500, and 1000 molecules cm$^{-3}$ s$^{-1}$ respectively. There is no error growth over time. Among the four representative species, the RRMS is highest for NO$_2$, ranging from 1.0% to 2.0% for $\delta$ ranging from $10^2$ to $10^3$ molecules cm$^{-3}$ s$^{-1}$. . For OH, ozone and sulfate, the RRMSs are below 0.3% in call cases. Figure 6 displays the spatial distribution of the relative error on the last day of the 2-year simulation, using a rate threshold $\delta$ of 500 molecules cm$^{-3}$ s$^{-1}$ as an example. The relative errors are below 0.5% everywhere for O$_3$, OH, and sulfate. The error for NO$_2$ reaches 1-10% at high latitudes, but this is still well within other systematic sources of errors in estimating NO$_2$ concentrations (Silvern et al., 2018). Results for rate thresholds $\delta$ of 100 and 1000 molecules cm$^{-3}$ s$^{-1}$ can be found in Figure S8-9.

[Figure]

**Figure 5**. Accuracy of the adaptive reduced chemistry mechanism algorithm over a two-year GEOS-Chem simulation (see text). The accuracy is measured by the 24-hour mean RRMS error on the end day of each month relative to a simulation including the full chemical mechanism. Rate thresholds $\delta$ of (a) 100, (b) 500 and (c) 1000 molecules cm$^{-3}$ s$^{-1}$ are used to partition the fast and slow species in the reduced mechanism. Results are shown for the median RRMS across all 228 species of the full mechanism and more specifically for ozone, OH, NO$_2$, and sulfate.

*10) Figure 4. Can the figure caption give more information here? What actually is being compared? Is this the RRMS in the monthly mean fields, or in the hourly values averaged to a monthly mean? Is this all of the species in the Jacobian?*

**Response**. Now we say.

Line 465. The accuracy is measured **by the 24-hour mean RRMS** error relative to a simulation including the full chemical mechanism **on the end day of each month**.

Line 467. Results are shown for the median RRMS across all 228 species of the full mechanism and more specifically for ozone, OH, NO$_2$, and sulfate.

*11) It's not obvious that the code for the annealing algorithm is included in the repository. I've had a look but can't find it.*
**Response**. We have uploaded the code. Please check.

*12) Conclusions. a. It would be useful to discuss whether this algorithm could be used within the adjoint framework for data assimilations, inversion studies? b. The authors discuss the suitability of this approach to minor mechanistic changes. However, if the algorithm is to be useful it needs to be sustainable within the software lifecycle of the chemistry transport model. Could this be spelt out in more detail? Presumably, if a new species was added the training algorithm (which species into which block and how many blocks etc) would need to the re-run with new data, but a small change in species lifetime would not lead to a re-running. It would be useful to have the conditions which are required for the training to be updated to be described.*
**Response.** Now we say this.
Line 294. (4) It is robust against small mechanistic changes, as these may not alter the choice of chemical regimes or may be accommodated by minor tweaking of the **regimes (new species may be assigned to their most appropriate groups on the basis of chemical logic)**. (5) It is robust against increases in model resolution, where source gridboxes (e.g., urban areas) will simply default to the full mechanism. **(6) If an adjoint is available for the full chemical solver, then it can also be used in our method since the software code of the full chemical solver (e.g. KPP) is retained.**

**Reviewer 2**

*Shen et al describes the implementation of a method of reducing the computational complexity of solving a chemical mechanism within GEOS-chem. The paper is interesting, although further revisions are required before it can be considered for publication.*

**Response**. Thanks for raising these good points. This feedback has significantly improved the manuscript.

*Major Comments*

*I found the discussion in Section 3.2 very hard to follow, specifically how blocks are grouped into regimes and then the subsequent changing of blocks from slow to fast if a gridbox does not correspond to any of the regimes. The sentence in question is*

*"Gridboxes that do not correspond to any of the M regimes need to be matched to one of the M regimes by moving some blocks from slow to fast, which will change the values of the corresponding indicators $y_{i,j}$ from 0 to 1."*

*Could the authors explain just how the mapping of species to blocks to regimes to these re-matched regimes is done? A diagram or pseudocode would be useful here. This crucial step is not explained well, and I'm not sure if this step is done online or not. How is the regime determined during a model run, and how is it ensured that the regimes are correctly matched (and what happens when they do not match)? This information is required to adequately understand the method presented.*

**Response**. Thanks. This process is done offline. The $20^{th}$ chemical regime is the full chemical mechanism, so every gridbox can be matched with a regime after some moves. We have included a diagram in the supplement and mentioned this in text.

Line 194. We check each of the *M* regimes and select the one that needs least number of moves from slow to fast, and this selection can be pre-defined so it does not add extra computational time. The $20^{th}$ chemical regime is the full mechanism, so every gridbox can be matched by the M regimes.

Line 199. A diagram for this process can be found in Figure S1.

[Figure]

Diagram for calculating the cost function

Concentrations of species in gridbox $j$, denoted as $\boldsymbol{n}_j = \{ n_{i,j},\ i$ is for species$\}$

Convert to $\boldsymbol{y}_j = \{ y_{i,j} \}$, where $y_{i,j}=1$ if the block is fast or $y_{i,j}=0$ if the block is slow. A fast block means at least one species in the block is fast

If $\boldsymbol{y}_j$ can be represented by the 20 chemical regimes

If $\boldsymbol{y}_j$ cannot be represented by the 20 chemical regimes

The number of fast species is calculated as $\sum_i y_{i,j}$

For each chemical regime, check if $\boldsymbol{y}_j$ can be represented by this chemical regime after moving some blocks from slow to fast. If it can, calculate the number of moves needed. We refer to $y^*_{i,j}$ as the indicators adjusted by these changes.

Identify the chemical regime that needs the least number of moves, and the number of fast species is calculated as $\sum_i y^*_{i,j}$

**Figure S1**. The diagram for calculating the cost function $Z_2$. More details can be found in text.

*I would be interested to know how robust the particular organisation determined from the Simulated Annealing algorithm is. Were multiple simulated annealing simulations performed? Was the rate of reduction of "temperature" changed to see if this affected the results? As with any global optimisation technique it is possible to get stuck in local minima, and a single run-through this algorithm will not be sufficient to determine whether the true minima has been found.*

**Response**. Thanks for raising this good point. In this study, we have run the optimization multiple times and also tried different temperature parameters. We present the one with lowest cost function. Now we make this clear in text.

Line 186. Throughout this study, we present the results with lowest cost function after running the optimization multiple times and using different temperature parameters.

The set of species blocks are not always exactly the same due to the random processes in simulated annealing. But in general, they can separate the oxidants, anthropogenic VOCs, and biogenic VOCs. Now we say.

Line 216. This algorithm still has shortcomings. There are some unexpected groupings (such as sulfur species and peroxyacetylnitrate) and separations (such as $HO_2$ and $H_2O_2$). The blocks are constructed by minimizing the number of fast species in the optimization, so species tend to be in the same block as long as they are fast or slow simultaneously. For example, isoprene products and CFCs are both slow in the stratosphere and clean regions, so they may be assigned into the same group (e.g., block 6). **In addition, there are still noticeable changes of species groups if we run the simulated annealing algorithm with different initializations and choices of the temperature parameter, even though the optimized blocks can generally separate the oxidants, anthropogenic VOCs, and biogenic VOCs (Table S1). Here we**

**chose the set of groupings that minimized the cost function for a number of realizations of the algorithm. These two shortcomings may be addressed by introducing regularization terms in the cost function to enforce known species relationships, which will implement this in a follow-up study.**

*While not essential for this manuscript, I would be interested to know if this classification has any load-balancing implications. I can imagine that for codes with MPI parallelisation across many nodes of a HPC, this method will increase the imbalance between different MPI tasks (while still decreasing the overall run-time). This could then lead to further speed improvements if the load-balancing is improved.*
**Response**. Now we have a paragraph to discuss this problem.

Line 309. The performance tests presented here were for a single-node implementation of GEOS-Chem using 12 CPUs in a shared-memory Open Message Passing (Open-MP) parallel environment. High-performance GEOS-Chem (GCHP) simulations can also be conducted in massively parallel environments with Message Passing Interface (MPI) communication between nodes and domain decomposition across nodes by groups of columns (Eastham et al., 2018). In principle, the chemical operator scales perfectly across nodes because it does not need to exchange information between columns (Long et al., 2015). However, differences in computational costs between columns (due to differences in chemical regimes) could result in load imbalance between nodes, degrading performance. In the current implementation of GCHP, the MPI domain decomposition is by clustered geographical columns in order to minimize exchange of information across nodes in the advection operator (Eastham et al., 2018). Such a decomposition would penalize our approach since different geographical domains may have different computational loads for chemistry (e.g., oceanic vs. continental regions). This could be corrected by using different MPI domain decompositions for different model operators, and tailoring the domain decomposition for the chemical operator to balance the number of fast species across nodes. Such an approach is used for example in the NCAR Community Earth System Model (CESM) where different domain decompositions are done for advection (clustered geographical regions) and for radiation (number of daytime columns).

*Most of the discussion and plots presented use a $\delta$ of 100 molecules cm$^{-3}$ s$^{-1}$ (or a range is presented), except when $\delta = 500$ is used for Figures 4 (the equivalent plot for $\delta = 100$ is Figure S5) and 5 and the discussion surrounding the 2-year runs in Section 4. Given that the $\delta = 100$ results seem noticeably better, why were the $\delta = 500$ presented in the main text? Are there equivalent plots (especially the Figure 5 equivalents) for the other values of $\delta$ used (100,1000)?*
**Response**. Now we show the results for rate thresholds $\delta$ of 100, 500 and 1000 molecules cm$^{-3}$ s$^{-1}$. The user can decide which to use based on their needs. Figure 5 is also updated.
Line 262. The best range for $\delta$ is between 100 and 1000 molecules cm$^{-3}$ s$^{-1}$, where the median RRMS error is below 1% and the improvement in computational performance is in the 30-40% range.

Line 271. Figure 5 shows the time evolution over two years of simulation of the median RRMS error for all species and also for the selected species OH, ozone, sulfate, and NO$_2$. The median RRMS for all species is 0.2%, 0.5%, and 0.8% for rate thresholds $\delta$ of 100, 500, and 1000 molecules cm$^{-3}$ s$^{-1}$ respectively. There is no error growth over time. Among the four representative species, the RRMS is highest for NO$_2$, ranging from 1.0% to 2.0% for $\delta$ ranging from $10^2$ to $10^3$ molecules cm$^{-3}$ s$^{-1}$. . For OH, ozone and sulfate, the RRMSs are below 0.3% in call cases. Figure 6 displays the spatial distribution of the relative error on the last day of the 2-year simulation, using a rate threshold $\delta$ of 500 molecules cm$^{-3}$ s$^{-1}$ as an example. The relative errors are below 0.5% everywhere for O$_3$, OH, and sulfate. The error for NO$_2$ reaches 1-10% at high latitudes, but this is still well within other systematic sources of errors in estimating NO$_2$

concentrations (Silvern et al., 2018). Results for rate thresholds $\delta$ of 100 and 1000 molecules cm$^{-3}$ s$^{-1}$ can be found in Figure S8-9.

[Figure]

**Figure 5**. Accuracy of the adaptive reduced chemistry mechanism algorithm over a two-year GEOS-Chem simulation (see text). The accuracy is measured by the 24-hour mean RRMS error on the end day of each month relative to a simulation including the full chemical mechanism. Rate thresholds $\delta$ of (a) 100, (b) 500 and (c) 1000 molecules cm$^{-3}$ s$^{-1}$ are used to partition the fast and slow species in the reduced mechanism. Results are shown for the median RRMS across all 228 species of the full mechanism and more specifically for ozone, OH, NO$_2$, and sulfate.

*Given the errors associated with halogen species presented in Figure S4, would there be a large drop in performance if these species were always treated as fast?*
**Response**. The test shows this will bring 4% more computation cost. Now we say
Line 155. This increases the computation cost of chemical integration by only 4% relative to letting the algorithm set them as either fast or slow.

*Minor Corrections*
*Page 6, Line 165: ",fast blocks,"*
**Response**. Fixed, thanks.

*Page 7, Equation 7: There is no D1, both $\Sigma$ are labelled with D2*
**Response**. Fixed, thanks.

*Page 16, Figure 1: The X-axes for the panels are slightly off-set. This can be clearly seen in the downward dotted lines.*
**Response**. Fixed, thanks.

[revised manuscript text omitted]
Here we refer to $y^*_{i,j}$ as the indicators adjusted by these changes. Thus, the fraction $Z_2$ of species that needs to be treated as
fast over the global domain is given by:

$$Z_2 = \frac{1}{\Omega} (\sum_{D_1} \sum_i y_{i,j} + \sum_{D_2} \sum_i y^*_{i,j}) \quad (6)$$

where $D_1$ are the gridboxes that can be represented by the top $M$ chemical regimes, and $D_2$ are the gridboxes that are
represented by other regimes and must be matched to the top $M$ regimes. A diagram for this process can be found in Figure
S1.

We tested a range of values from 3 to 20 for the number $N$ of blocks. In this testing we used a threshold $\delta = 100$ molecules cm$^{-3}$ s$^{-1}$ to partition fast and slow species, following Santillana et al. (2010), and a number $M = 20$ of chemical regimes (see next paragraph for choice of $M$). Figure 1 shows the fraction of fast species in the global domain ($Z_2$) as a function of $N$. If $N$ is low such that blocks are large, there is more likelihood that a species in a given block will be fast causing all species in the block to be treated as fast. If $N$ is high, more blocks will need to be moved from slow to fast in order to match the limited number $M$ of chemical regimes. For $M = 20$ we thus find an optimal value $N = 12$ at which only 40% of the species need to be treated as fast.

Table 1 lists  the species of these 12 blocks . Oxidants such as OH, O$_3$, and NO$_2$ are important under all circumstances so block 8 and 9 are fast in most gridboxes. Nonmethane VOCs species often have low concentrations outside of the continental boundary layer, and very low concentrations in the stratosphere, so the dominant VOC blocks 1-7 are fast in fewer than 40% of gridboxes. Anthropogenic VOC species (blocks 4 and 5) are found to be fast in boundary layer and daytime mid-troposphere (Figure S2-3). Biogenic VOC species have shorter lifetimes, so they are found to be fast only in lower and middle troposphere over the land (Figure S4-5).

This algorithm still has shortcomings. There are some unexpected groupings (such as sulfur species and peroxyacetylnitrate) and separations (such as HO$_2$ and H$_2$O$_2$). The blocks are constructed by minimizing the number of fast species in the optimization, so species tend to be in the same block as long as they are fast or slow simultaneously. For example, isoprene products and CFCs are both slow in the stratosphere and clean regions, so they may be assigned into the same group (e.g., block 6). In addition, there are still noticeable changes of species groups if we run the simulated annealing algorithm with different initializations and choices of the temperature parameter, even though the optimized blocks can generally separate the oxidants, anthropogenic VOCs, and biogenic VOCs (Table S1). Here we chose the set of groupings that minimized the cost function for a number of realizations of the algorithm. 
[revised manuscript text omitted]

---

## Referee Report (RR1)

**Re-review of Shen et al: An adaptive method for speeding up the numerical integration of chemical mechanisms in atmospheric chemistry models: application to GEOS-Chem version 12.0.0**

Many thanks to the authors for addressing mine and the other Referee's points. The changes have helped clarify matters greatly. I have a few further questions/suggestions before I can recommend publication.

**Major comments**

On the Simulated Annealing Algorithm used, I am concerned by the statement:

"In addition, there are still noticeable changes of species groups if we run the simulated annealing algorithm with different initializations and choices of the temperature parameter, even though the optimized blocks can generally separate the oxidants, anthropogenic VOCs, and biogenic VOCs (Table S1)."

To me, this means that the energy landscape is full of local minima and/or is possibly degenerate. Re-running the algorithm should give the same global minimum multiple times, if the algorithm is robust. Table S1 only lists two other potential groupings – why show only these two in the supplementary information? How many times was this algorithm run for each value of N? Given that when calculating N, the value of  $\delta$  is fixed at 100 molecules cm-3 s-1, but later simulations change  $\delta$  to higher values, how robust is the categorisation to the value of  $\delta$ ?

It may be that the groupings don't actually matter that much – if there are lots of local minima then each could give similar performance. However, this should ideally be tested if this is the case to see if the errors remain the same. Simulated Annealing is not a great global optimisation technique really, and others such as Basin Hopping or Genetic Algorithms have shown better performance for problems with a large number of potential solutions.

Diagram S1 should be placed in the main text. If I understand this method correctly, you use a training dataset from 4 GEOS-Chem simulations that have been run for 10-days and use output from every 6 hours. Using this you have categorised the species into 12 different distinct blocks (using simulated annealing), which are then combined together into 20 different regimes, and you have assigned each gridbox a regime. Please clarify further if this is not correct.

Does the regime that a gridbox has change in time during the simulation, and if so how? In your response you state that this is calculated offline, so how does time of day or season affect things? If the emissions were changed, would everything need to be re-calculated again? Similarly, if you are wanting to run a pre-industrial or future scenario, what would need to be changed or re-calculated? Figure 3 and S2-S5 show that things are changing in time, but I am not clear how this is determined given your response that this is calculated offline.

**Minor Comments**

Line 184: Here you state that "Among the N blocks, 3 are allocated to the reactive inorganic halogen species, and N-3 are allocated to the other species.", and on line 203 you state that "We tested a range of values from 3 to 20 for the number N of blocks". Does this mean that you just shuffled the halogen species between these three along with everything else, or were you testing 6 to 23 blocks?

**Line 190: "selected representative" and "and a full listing is in Fig. S2"**

Line 254: you use " $10^{2}$ " and " $10^{3}$ " here, but 100 and 1000 elsewhere.

The colourbars on figures S2-S5 are completely redundant.

---

## Author Response (AR2)

**Response to referee comments on "An adaptive method for speeding up the numerical integration of chemical mechanisms in atmospheric chemistry models: application to GEOS-Chem version 12.0.0"**

We thank the referees for their careful reading of the manuscript and the valuable comments. This document is organized as follows: the Referee's comments are in *italic*, our responses are in plain text, and all the revisions in the manuscript are shown in blue. **Boldface blue text** denotes text written in direct response to the Referee's comments. The line numbers in this document refer to the updated manuscript.

*Many thanks to the authors for addressing mine and the other Referee's points. The changes have helped clarify matters greatly. I have a few further questions/suggestions before I can recommend publication.*

**Major comments**

*On the Simulated Annealing Algorithm used, I am concerned by the statement:*

*"In addition, there are still noticeable changes of species groups if we run the simulated annealing algorithm with different initializations and choices of the temperature parameter, even though the optimized blocks can generally separate the oxidants, anthropogenic VOCs, and biogenic VOCs (Table S1)."*

*To me, this means that the energy landscape is full of local minima and/or is possibly degenerate. Re-running the algorithm should give the same global minimum multiple times, if the algorithm is robust. Table S1 only lists two other potential groupings – why show only these two in the supplementary information? How many times was this algorithm run for each value of N? Given that when calculating N, the value of $\delta$ is fixed at 100 molecules $cm^{-3} s^{-1}$, but later simulations change $\delta$ to higher values, how robust is the categorisation to the value of $\delta$?*

**Response**. We thank the reviewer for raising this good point. But finding the global minimum of our cost function is really expensive and also not necessary. Instead we run our algorithm for many different timesteps and we select the one has lowest cost function. As long as the optimized species blocks can help significantly reduce the computer time of the chemical integration, our method can still be very useful. We have a follow-up project to make the species blocks more stable by introducing a regularization term that defines the species' distances as learned from their reactant-product relationships. The preliminary result of the revised method is more chemically logical, but it still has some unresolved issues so we are not able to present it in this work.

Here we show more other groups and the results are consistent with what we have presented in the manuscript. They can generally separate the oxidants, anthropogenic VOCs and biogenic VOCs, even though there are noticeable changes of groupings.

Optimization 3

| 1 | CH2I2 CH2ICl LTRO2H LTRO2N SOAGX CH3I TOLU TRO2 OCS CHBr3 CH2Cl2 CHCl3 HCFC22 PP PIP HC187 ROH IEPOXOO PO2 R4N1 HCOOH GLYC |
|---|---|
| 2 | LBRO2H LBRO2N SO4H2 IMAE BENZ BRO2 RA3P RB3P CH3Cl RP EOH A3O2 HAC |
| 3 | SO4H1 PPN DMS PAN RCO3 MCO3 SO2 |

| | |
|---|---|
| 4 | CH2IBr ISN1OA ISN1OG LISOPNO3 LVOCOA LVOC PYAC SOAMG DHDN CH3CCl3 H1301 H2402 PMNN CCl4 CFC11 CFC12 CFC113 CFC114 CFC115 H1211 IEPOXD CH2Br2 HCFC123 HCFC141b HCFC142b CH3Br PRPN DHPCARP IAP HPC52O2 MOBA DHMOB ISNP MAOP MRP RIPD ETHLN ISNOHOO NPMN MOBAOO DIBOO LIMO ISNOOB INPN MACRNO2 ISOPNB MVKOO GAOO CH3CHOO MGLYOO PRN1 MGLOO MONITU MAN2 ISNOOA ISOPNDO2 MACROO MACRN MAOPO2 OLNN LIMO2 ISOPNBO2 ISOPND NMAO3 ISN1 HC5 INO2 |
| 5 | LISOPOH LXRO2H LXRO2N SOAIE SOAME DHDC MONITA IEPOXA IEPOXB XRO2 IMAO3 XYLE HPALD VRP HONIT RIPB RIPA MTPA MTPO IPMN MONITS MVKN CH2OO PROPNN ISOP OLND PIO2 HC5OO VRO2 RIO2 MRO2 MACR MVK |
| 6 | INDIOL IONITA N GLYX R4N2 PRPE |
| 7 | MSA MAP ETP SO4 ALK4 R4P C3H8 ATOOH C2H6 B3O2 ATO2 KO2 ACTA MGLY ACET ETO2 R4O2 RCHO MEK ALD2 |
| 8 | CO2 N2O HNO4 HNO2 MP H CH4 H2O2 CH2O CO NO O1D O |
| 9 | MPN N2O5 HNO3 MO2 O3 HO2 NO3 NO2 H2O OH |
| 10 | I2O2 BrNO2 Cl2O2 IONO OIO OClO HOI IONO2 Cl2 I IO BrO Br |
| 11 | AERI ISALA ISALC I2O4 I2O3 IBr INO HI ICl ClNO2 BrSALC BrSALA I2 |
| 12 | ClOO BrCl Br2 BrNO3 HOBr HOCl ClNO3 Cl HBr ClO HCl |

Optimization 4

| | |
|---|---|
| 1 | SO4H2 IMAE N EOH GLYX KO2 HAC |
| 2 | CH2I2 CH2ICl LBRO2H LBRO2N LTRO2H LTRO2N LXRO2H BENZ CH3I TOLU TRO2 BRO2 XRO2 CH2Cl2 RA3P XYLE RP A3O2 R4N1 |
| 3 | LISOPOH LXRO2N SOAGX SOAIE SOAME DHDC MONITA IEPOXA IEPOXB IMAO3 PP PIP HONIT MTPA MTPO IPMN ROH MONITS MONITU PO2 ISOP OLND PIO2 RIO2 MVK |
| 4 | SO4H1 PPN DMS PAN RCO3 ACTA ETO2 PRPE ALD2 MCO3 SO2 |
| 5 | CH2IBr ISN1OA ISN1OG PYAC SOAMG CH3CCl3 H1301 H2402 CCl4 CFC11 CFC12 CFC113 CFC114 CFC115 H1211 OCS CHBr3 CHCl3 CH2Br2 HCFC123 HCFC141b HCFC142b HCFC22 CH3Br NPMN NMAO3 |
| 6 | LISOPNO3 LVOCOA LVOC DHDN PMNN IEPOXD PRPN HPALD DHPCARP HC187 IAP VRP HPC52O2 MOBA DHMOB RIPB ISNP MAOP MRP RIPA RIPD ETHLN ISNOHOO MOBAOO DIBOO LIMO ISNOOB INPN MACRNO2 ISOPNB MVKOO GAOO CH3CHOO MGLYOO IEPOXOO MVKN PRN1 MGLOO CH2OO PROPNN MAN2 ISNOOA ISOPNDO2 MACROO MACRN MAOPO2 HCOOH OLNN LIMO2 ISOPNBO2 HC5OO ISOPND GLYC VRO2 ISN1 HC5 INO2 MRO2 MACR |
| 7 | INDIOL MSA IONITA MAP ETP RB3P CH3Cl SO4 ALK4 R4P C3H8 ATOOH C2H6 B3O2 ATO2 MGLY ACET R4O2 R4N2 RCHO MEK |
| 8 | CO2 N2O HNO4 HNO2 MP H CH4 H2O2 CH2O CO O1D O |
| 9 | MPN N2O5 HNO3 MO2 O3 NO HO2 NO3 NO2 H2O OH |
| 10 | AERI ISALA ISALC I2O4 I2O3 IBr INO HI ICl ClNO2 BrSALC I2 |
| 11 | ClOO BrCl Br2 BrNO3 HOBr HOCl ClNO3 Cl HBr ClO HCl |
| 12 | I2O2 BrNO2 Cl2O2 IONO OIO OClO HOI BrSALA IONO2 Cl2 I IO BrO Br |

We have production and loss rates for the 228 species in the first 10 days of February, May, August and November, sampled every 6 hours, which yields a matrix of 72(longitude) × 46(latitude) × 72(altitude) × 228(species) ×160(timesteps). It is very expensive to optimize on such a large matrix. In order to reduce the computational cost, we run the optimization using data for each timestep, and then we report the average cost function by applying the optimized blocks to all timesteps. Now we say this in the text.

Line 152. We use for this purpose a training dataset from a GEOS-Chem simulation for 2013, consisting of the global ensemble of tropospheric and stratospheric gridboxes for the first 10 days of February, May, August, and November sampled every 6 hours **(160 time steps in total). To reduce the computational cost, we optimize the partitioning of species into blocks for each individual timestep, resulting in 160 different partitionings, and we then select the partitioning that yields the lowest cost function when applied to all timesteps.**

Here I am showing the species blocks from the same optimization process but using a threshold ($\delta$) of 1000 molecules cm$^{-3}$ s$^{-1}$. They can generally separate the oxidants, anthropogenic VOCs and biogenic VOCs, even though there are also noticeable changes of species groups.

Optimization 1

| | |
|---|---|
| 1 | N2O,HNO2,MP,H,CH4,H2O2,O1D,O |
| 2 | INDIOL,PPN,IONITA,ALK4,R4P,GLYX,RCO3,KO2,R4O2,R4N2,PRPE,RCHO,MEK |
| 3 | LBRO2H,IMAE,DHDC,BENZ,BRO2,RA3P,RB3P,CH3Cl,RP,PP,SO4,PIP,C3H8,EOH,A3O2,PO2,B3O2,HAC |
| 4 | CH2I2,CH2ICl,CH2IBr,ISN1OA,ISN1OG,LBRO2N,LTRO2H,LTRO2N,LVOCOA,LVOC,LXRO2H,LXRO2N,PYAC,SOAGX,SOAME,SOAMG,DHDN,CH3CCl3,CH3I,H1301,H2402,PMNN,TOLU,CCl4,CFC11,CFC12,CFC113,CFC114,CFC115,H1211,IEPOXD,TRO2,N,OCS,XRO2,CHBr3,CH2Cl2,CHCl3,CH2Br2,HCFC123,HCFC141b,HCFC142b,HCFC22,XYLE,CH3Br,PRPN,IAP,VRP,HPC52O2,MOBA,HONIT,DHMOB,ISNP,MAOP,MRP,RIPD,ETHLN,ISNOHOO,NPMN,MOBAOO,DIBOO,LIMO,ISNOOB,MACRNO2,ROH,ISOPNB,MVKOO,GAOO,CH3CHOO,MGLYOO,MVKN,MGLOO,MONITU,PROPNN,MAN2,ISNOOA,ISOPNDO2,MACROO,R4N1,MACRN,MAOPO2,LIMO2,ISOPNBO2,ISOPND,NMAO3 |
| 5 | LISOPOH,LISOPNO3,SO4H2,SOAIE,MONITA,IEPOXA,IEPOXB,IMAO3,HPALD,DHPCARP,HC187,RIPB,RIPA,MTPA,MTPO,IPMN,INPN,MONITS,IEPOXOO,PRN1,CH2OO,ISOP,HCOOH,OLND,OLNN,PIO2,HC5OO,GLYC,VRO2,ISN1,HC5,RIO2,INO2,MRO2,MACR,MVK |
| 6 | MSA,SO4H1,MAP,ETP,ATOOH,C2H6,ATO2,ACTA,MGLY,ETO2,ALD2 |
| 7 | DMS,PAN,ACET,MCO3,SO2 |
| 8 | CO2,HNO4,CH2O,CO,NO,HO2,OH |
| 9 | MPN,N2O5,HNO3,MO2,O3,NO3,NO2,H2O |
| 10 | AERI,ISALA,ISALC,I2O4,I2O2,I2O3,IBr,INO,HI,ICl,Cl2O2,IONO,ClNO2,BrSALC,BrSALA,I2,Cl2 |
| 11 | BrNO2,OIO,OClO,BrCl,HOI,Br2,IONO2,BrNO3,I,IO,HOBr,HOCl,ClNO3,HBr,HCl |
| 12 | ClOO,BrO,Br,Cl,ClO |

Optimization 2

| | |
|---|---|
| 1 | CH2I2,CH2ICl,CH2IBr,ISN1OA,ISN1OG,LVOCOA,LVOC,PYAC,SOAME,SOAMG,DHDN,CH3CC |

| | |
|---|---|
| | l3,CH3I,H1301,H2402,PMNN,CCl4,CFC11,CFC12,CFC113,CFC114,CFC115,H1211,N,OCS,CHBr3, CH2Cl2,CHCl3,CH2Br2,HCFC123,HCFC141b,HCFC142b,HCFC22,CH3Br,PRPN,IAP,MOBA,ISNP, MAOP,MRP,ETHLN,NPMN,MOBAOO,DIBOO,ISNOOB,MACRNO2,MVKOO,GAOO,MGLYOO, MAN2,ISNOOA,ISOPNDO2,MACROO,MACRN,MAOPO2,NMAO3 |
| 2 | SO4H1,N2O,DMS,SO2 |
| 3 | LBRO2N,LISOPOH,LISOPNO3,LTRO2H,LTRO2N,SOAGX,SOAIE,DHDC,TOLU,TRO2,IEPOXA,I EPOXB,PIP,HPALD,HC187,RIPB,RIPA,INPN,IEPOXOO,GLYX,CH2OO,ISOP,HCOOH,PIO2,HC5 OO,GLYC,VRO2,ISN1,RIO2,INO2,MRO2,MACR,MVK |
| 4 | PPN,RB3P,CH3Cl,SO4,ALK4,R4P,C3H8,ATOOH,C2H6,B3O2,RCO3,MGLY,R4O2,RCHO,MEK |
| 5 | INDIOL,LBRO2H,IMAE,BENZ,BRO2,IONITA,IMAO3,RA3P,RP,PP,EOH,MTPA,MTPO,IPMN,A3 O2,PO2,OLND,KO2,R4N2,HAC,PRPE |
| 6 | LXRO2H,LXRO2N,SO4H2,MONITA,IEPOXD,XRO2,XYLE,DHPCARP,VRP,HPC52O2,HONIT,DH MOB,RIPD,ISNOHOO,LIMO,ROH,MONITS,ISOPNB,CH3CHOO,MVKN,PRN1,MGLOO,MONITU ,PROPNN,R4N1,OLNN,LIMO2,ISOPNBO2,ISOPND,HC5 |
| 7 | MSA,MAP,ETP,ATO2,ACTA,ACET,ETO2,ALD2 |
| 8 | CO2,HNO4,HNO2,PAN,MP,H,CH4,H2O2,MCO3,CO,O1D,O |
| 9 | MPN,N2O5,HNO3,CH2O,MO2,O3,NO,HO2,NO3,NO2,H2O,OH |
| 10 | BrNO2,OIO,OClO,BrCl,HOI,Br2,IONO2,BrNO3,I,IO,HOBr,HOCl,ClNO3,BrO,Br,HBr,ClO,HCl |
| 11 | AERI,ISALA,ISALC,I2O4,I2O2,I2O3,IBr,INO,HI,ICl,Cl2O2,IONO,ClNO2,BrSALC,BrSALA,I2,Cl2 |
| 12 | ClOO,Cl |

Optimization 3

| | |
|---|---|
| 1 | MSA,SO4H1,MAP,SO4,ATO2,ACTA,ACET,ETO2,ALD2 |
| 2 | CH2I2,CH2ICl,CH2IBr,ISN1OA,ISN1OG,LVOCOA,LVOC,PYAC,SOAME,SOAMG,DHDN,CH3C Cl3,MONITA,CH3I,H1301,H2402,PMNN,CCl4,CFC11,CFC12,CFC113,CFC114,CFC115,H1211,IE POXD,N,OCS,CHBr3,CH2Cl2,CHCl3,CH2Br2,HCFC123,HCFC141b,HCFC142b,HCFC22,CH3Br, PRPN,DHPCARP,IAP,VRP,MOBA,HONIT,ISNP,MAOP,MRP,RIPD,ETHLN,ISNOHOO,NPMN,M OBAOO,DIBOO,LIMO,ISNOOB,MACRNO2,ROH,MONITS,ISOPNB,MVKOO,GAOO,CH3CHO O,MGLYOO,MVKN,PRN1,MGLOO,MONITU,CH2OO,PROPNN,MAN2,ISNOOA,ISOPNDO2,M ACROO,MACRN,MAOPO2,OLNN,LIMO2,ISOPNBO2,ISOPND,NMAO3 |
| 3 | LISOPOH,LISOPNO3,LXRO2H,LXRO2N,SOAIE,IEPOXA,IEPOXB,XRO2,IMAO3,XYLE,HPAL D,HPC52O2,DHMOB,RIPB,RIPA,IPMN,INPN,IEPOXOO,R4N1,ISOP,HC5OO,VRO2,ISN1,HC5,R IO2,INO2,MRO2,MACR,MVK |
| 4 | PPN,ETP,CH3Cl,ALK4,R4P,ATOOH,C2H6,RCO3,KO2,MGLY,R4O2,RCHO,MEK |
| 5 | INDIOL,SOAGX,DHDC,IONITA,PIP,MTPA,MTPO,GLYX,PO2,HCOOH,OLND,PIO2,GLYC,R4N 2,PRPE |
| 6 | N2O,DMS,PAN,MP,H2O2,MCO3,SO2 |
| 7 | LBRO2H,LBRO2N,LTRO2H,LTRO2N,SO4H2,IMAE,BENZ,TOLU,TRO2,BRO2,RA3P,RB3P,RP, PP,C3H8,HC187,EOH,A3O2,B3O2,HAC |
| 8 | CO2,HNO4,HNO2,H,CH4,CH2O,CO,NO,HO2,O1D,OH,O |
| 9 | MPN,N2O5,HNO3,MO2,O3,NO3,NO2,H2O |
| 10 | AERI,ISALA,ISALC,I2O4,I2O2,I2O3,IBr,INO,HI,ICl,Cl2O2,IONO,ClNO2,BrSALC,BrSALA,I2,Cl 2 |

| 11 | BrNO2,OIO,OClO,BrCl,HOI,Br2,IONO2,BrNO3,I,IO,HOBr,HOCl,ClNO3,BrO,Br,HBr,ClO,HCl |
| 12 | ClOO,Cl |

*It may be that the groupings don't actually matter that much – if there are lots of local minima then each could give similar performance. However, this should ideally be tested if this is the case to see if the errors remain the same. Simulated Annealing is not a great global optimisation technique really, and others such as Basin Hopping or Genetic Algorithms have shown better performance for problems with a large number of potential solutions.*

**Response**. Thanks for these suggestions. Yes, the error remains very similar if we use different species groups. We have tested the error for the two species blocks in Table S1. The median RRMS errors are 0.67% and 0.63%, compared to 0.59% if using the species blocks in the main text. We have added these two numbers in the Table S1 and we also say this in the text.

Line 263. Running the optimizing algorithm may produce different groupings of species (e.g. Table S1), but they show similar errors.

Thanks for suggesting these two optimization algorithms. We would like to try them in the following-up project.

*Diagram S1 should be placed in the main text. If I understand this method correctly, you use a training dataset from 4 GEOS-Chem simulations that have been run for 10-days and use output from every 6 hours. Using this you have categorised the species into 12 different distinct blocks (using simulated annealing), which are then combined together into 20 different regimes, and you have assigned each gridbox a regime. Please clarify further if this is not correct.*

**Response**. We have moved Diagram S1 to Figure 1. The gridbox will pick the most appropriate chemical regime at the beginning of each timestep. We have said this in the abstract and also added more discussion in the text.

Abstract. We do this by constructing a limited set of reduced chemical mechanisms (chemical regimes) to cover the range of atmospheric conditions, and then pick locally and on the fly which mechanism to use for a given gridbox and time step on the basis of computed production and loss rates for individual species.

Line 184. At the beginning of each timestep, we pick the chemical regime to use for each gridbox on the basis of computed production and loss rates for individual species

*Does the regime that a gridbox has change in time during the simulation, and if so how? In your response you state that this is calculated offline, so how does time of day or season affect things? If the emissions were changed, would everything need to be re-calculated again? Similarly, if you are wanting to run a pre-industrial or future scenario, what would need to be changed or re-calculated? Figure 3 and S2-S5 show that things are changing in time, but I am not clear how this is determined given your response that this is calculated offline.*

**Response**. Thanks for this careful reading. The chemical regime used by a gridbox is not constant and it is updated on the fly based on the species production and loss rates. Hence, our algorithm can adapt to any changes in time of day, seasons and anthropogenic emission levels. The 'offline calculation' in our last

reply actually referred to how we match other chemical regimes to the top $M$ regimes. Please check our new text.

Line 175. Gridboxes that do not correspond to any of the $M$ regimes need to be matched to one of the $M$ regimes by moving some blocks from slow to fast, which will change the values of the corresponding indicators $y_{i,j}$ from 0 to 1. We check each of the $M$ regimes and select the one that needs the least number of moves from slow to fast, and this selection can be pre-defined so it does not add extra computational time.

**Minor Comments**

Line 184: Here you state that "Among the N blocks, 3 are allocated to the reactive inorganic halogen species, and N-3 are allocated to the other species.", and on line 203 you state that "We tested a range of values from 3 to 20 for the number N of blocks". Does this mean that you just shuffled the halogen species between these three along with everything else, or were you testing 6 to 23 blocks?

**Response**. Sorry, this is a typo. We test N from 5 to 20.

Line 190: "selected representative" and "and a full listing is in Fig. S2" Line 254: you use "$10^2$" and "$10^3$" here, but 100 and 1000 elsewhere. The colourbars on figures S2-S5 are completely redundant.

**Response**. Thanks. We deleted "selected representative". Now we use 100 and 1000. We have removed the colorbar in Figure S2-S5.

[revised manuscript text omitted]

Diagram for calculating the cost function $Z_2$

Concentrations of species in gridbox $j$, denoted as $\boldsymbol{n}_j = \{ n_{i,j}, i$ is for species$\}$

Convert to $\boldsymbol{y}_j = \{ y_{i,j} \}$, where $y_{i,j}=1$ if the block is fast or $y_{i,j}=0$ if the block is slow. A fast block means at least one species in the block is fast.

If $\boldsymbol{y}_j$ can be represented by the $M$ chemical regimes

If $\boldsymbol{y}_j$ cannot be represented by the $M$ chemical regimes

The number of fast species is calculated as $\sum_i y_{i,j}$

For each of the $M$ chemical regimes, check if $\boldsymbol{y}_j$ can be represented by this chemical regime after moving some blocks from slow to fast. If it can, calculate the number of moves needed. We refer to $y^*_{i,j}$ as the indicators adjusted by these changes.

Identify the chemical regime that needs the least number of moves, and the number of fast species is calculated as $\sum_i y^*_{i,j}$

420

**Figure 1**. The diagram for calculating the cost function $Z_2$.

[Figure]

**Figure 2**. Minimum of cost function $Z_2$ (global fraction of chemical species treated as fast) as a function of the number $N$ of
425    blocks used to group the species for mechanism reduction. Values were computed using the GEOS-Chem troposphere +
stratosphere simulation on the first days of February, April, August and November 2013, over 24 hours and sampled every 6
hours. Shaded area shows the standard deviation of the cost function minimum computed for each sample.

430

[Figure]

**Figure 3**. Speed-up of the chemical computation as a function of the number $M$ of chemical mechanism subsets (chemical regimes) used in the coupled implicit solver of the GEOS-Chem model for adaptive simulation of the troposphere and stratosphere. Top: Minimum of cost function $Z_2$ (global fraction of chemical species treated as fast) as a function of the number of chemical regimes. Bottom: Percentage of model gridboxes that can be represented by the $M$ chemical regimes without adjustment (see Equation 5 and related text). Dashed lines show the values for $M = 20$. For both panels, results are for the first 10 days of February, May, August, and November sampled every 6 hours (shaded area denotes one standard deviation of results sampled every 6 hours).

[Figure]

Figure 4. Chemical mechanism complexity needed in different regions of the atmosphere. The Figure identifies the chemical regime from Table 2 needed to simulate a given GEOS-Chem gridbox on August 1 2013 at 0 and 12 GMT. The percentage of the 228 species treated as fast (requiring coupled implicit solution) in that chemical regime is shown on the colorbar and more details are in Tables 1 and 2. Results are shown for different altitudes and using a threshold $\delta$ of 100 molecules $cm^{-3}$ $s^{-1}$.

[Figure]

445 **Figure 5**. Performance and accuracy of the adaptive chemical mechanism reduction method for different rate thresholds δ (molecules cm[-3] s[-1]) to separate fast and slow species. The performance is measured by the reduction in computing processor unit (CPU) time for the chemical operator, and the accuracy is measured by the median relative root mean square (RRMS) error for species concentrations relative to a global GEOS-Chem simulation for the troposphere and stratosphere using the full chemical mechanism (228 species treated as fast). The second *x* axis gives the global fraction of species that need to be

450 treated as fast depending on the value of δ. The number of blocks (*N*) is 12 and the number of chemical regimes (*M*) is 20.

[Figure]

**Figure 6**. Accuracy of the adaptive reduced chemistry mechanism algorithm over a two-year GEOS-Chem simulation (see text). The accuracy is measured by the 24-hour mean RRMS error on the end day of each month relative to a simulation including the full chemical mechanism. Rate thresholds $\delta$ of (a) 100, (b) 500 and (c) 1000 molecules $cm^{-3}$ $s^{-1}$ are used to partition the fast and slow species in the reduced mechanism. Results are shown for the median RRMS across all 228 species of the full mechanism and more specifically for ozone, OH, $NO_2$, and sulfate.

Relative error in the adpative mechanism reduction method

**Figure 7**. Relative error from the adaptive mechanism reduction method after two years of simulation in the GEOS-Chem global 3-D model for tropospheric-stratospheric chemistry. The figure shows relative differences of 24-h average OH, ozone, sulfate and $NO_2$ concentrations relative to the full-chemistry simulation on the last day of the two-year simulation (2013-2014). The relative error for surface $NO_2$ can be up to ±10% in polar regions. The calculation uses a rate threshold $\delta$ = 500 molecules $cm^{-3}$ $s^{-1}$ to partition the species between fast and slow. The number of blocks ($N$) is 12 and the number of chemical regimes ($M$) is 20.